# Direct RNA sequencing on nanopore arrays redefines the transcriptional complexity of a viral pathogen

Daniel P. Depledge [1], Kalanghad Puthankalam Srinivas [1], Tomohiko Sadaoka [2], Devin Bready [3], Yasuko Mori [2], Dimitris G. Placantonakis [3,4,5,6,7], Ian Mohr[1,5] & Angus C. Wilson [1,5]

Characterizing complex viral transcriptomes by conventional RNA sequencing approaches is complicated by high gene density, overlapping reading frames, and complex splicing patterns. Direct RNA sequencing (direct RNA-seq) using nanopore arrays offers an exciting alternative whereby individual polyadenylated RNAs are sequenced directly, without the recoding and amplification biases inherent to other sequencing methodologies. Here we use direct RNA-seq to profile the herpes simplex virus type 1 (HSV-1) transcriptome during productive infection of primary cells. We show how direct RNA-seq data can be used to define transcription initiation and RNA cleavage sites associated with all polyadenylated viral RNAs and demonstrate that low level read-through transcription produces a novel class of chimeric HSV-1 transcripts, including a functional mRNA encoding a fusion of the viral E3 ubiquitin ligase ICP0 and viral membrane glycoprotein L. Thus, direct RNA-seq offers a powerful method to characterize the changing transcriptional landscape of viruses with complex genomes.

[1] Department of Microbiology, New York University School of Medicine, New York, NY 10016, USA. [2] Division of Clinical Virology, Center for Infectious Diseases, Kobe University Graduate School of Medicine, 7-5-1 Kusunoki-cho, Chuo-ku, Kobe 650-0017, Japan. [3] Department of Neurosurgery, New York University School of Medicine, New York, NY 10016, USA. [4] Kimmel Center for Stem Cell Biology, New York University School of Medicine, New York, NY 10016, USA. [5] Laura and Isaac Perlmutter Cancer Center, New York University School of Medicine, New York, NY 10016, USA. [6] Brain Tumor Center, New York University School of Medicine, New York, NY 10016, USA. [7] Neuroscience Institute, New York University School of Medicine, New York, NY 10016, USA. Correspondence and requests for materials should be addressed to D.P.D. (email: daniel.depledge@nyulangone.org) or to A.C.W. (email: angus.wilson@nyulangone.org)

Herpesviruses are adept viral pathogens that have co-evolved with their hosts over millions of years. Like all viruses, their success is predicated on repurposing of the host transcriptional and translational machinery[1,2], and through the use of compact, gene-dense genomes with exceptional coding potential[3–7]. The 152-kb double-stranded DNA genome of herpes simplex virus type 1 (HSV-1) includes at least 80 distinct poly-adenylated transcripts. These predominantly encode single-exon open-reading frames (ORFs), some transcribed as polycistronic mRNAs, along with a smaller number of noncoding RNAs[8,9]. These are traditionally grouped into three kinetic classes termed immediate-early, early, and late[10–12]. Although splicing of HSV-1 RNAs is infrequent, exceptions include RNAs encoding ICP0, ICP22, UL15p, and ICP47, as well as the noncoding latency-associated transcript (LAT).

Conventional RNA-sequencing methodologies, while highly reproducible, utilize multiple recoding steps (e.g., reverse transcription, second-strand synthesis, and in some cases, PCR amplification) during library preparation that may introduce errors or bias in the resulting sequence data[13]. Data quality may be further convoluted by the use of short-read sequencing technologies, which require well-curated reference genomes to accurately assess the abundance and complexity of transcription in a given system. Loss of information on transcript isoform diversity, including splice variants, is especially problematic[14]. Despite these inherent difficulties, recent studies have shown that host transcription and mRNA processing are extensively remodeled during HSV-1 infection[15–17], and recent studies using cDNA-based short- and long-read sequencing technologies indicate that the HSV-1 transcriptome, like other herpesviruses[6,18], may be substantially more complex than previously recognized[19–21].

To examine this in detail, we have employed a new methodology for direct single-molecule sequencing of native poly-adenylated RNAs using nanopore arrays[22]. Specifically, we have used the Oxford Nanopore Technologies MinION platform to directly sequence polyadenylated host and viral RNAs from infected human primary fibroblasts at early and late stages of infection. Error correction, a prerequisite for current nanopore sequence-read datasets, and the generation of pseudotranscripts are guided using Illumina short-read sequence data from the same source material.

We begin by highlighting the fidelity and reproducibility of direct RNA-seq, while also leveraging short-read Illumina sequencing data to enable a new approach to error correction that significantly increases the proportion of error-free transcript sequences from which internal ORFs can be accurately translated to predict protein sequences. Using the polyadenylated fraction of the HSV-1 transcriptome, we define multiple new transcription initiation sites that produce mRNAs encoding novel or alternative ORFs. We provide evidence for read-through of polyadenylation signals in a number of HSV-1 transcription units to produce a new class of spliced transcripts with the potential to encode novel protein fusions. Finally, we show that one of these, a fusion between the ORFs encoding the viral E3 ubiquitin ligase ICP0 and viral membrane glycoprotein L, produces a 32-kDa polypeptide expressed with late kinetics. Taken together, this study demonstrates the power of direct RNA-seq to annotate complex viral transcriptomes and to identify novel polyadenylated RNA isoforms that further expand the coding potential of gene-dense viral genomes.

## Results

**Nanopore sequencing of host and viral transcriptomes.** To evaluate the reproducibility of direct RNA sequencing using nanopore arrays, total RNA was prepared from two biological replicates of normal human dermal fibroblasts (NHDF) infected with HSV-1 GFP-Us11 strain Patton (hereafter HSV-1 Patton)[23,24] for 18 h. Sequencing libraries were generated from the poly(A)+ RNA fraction and sequenced on a MinION MkIb with R9.4 flow cell with a run time of 18 h, yielding ~380,000 (replicate #1) and 218,000 (replicate #2) nanopore sequence reads (Fig. 1a, Supplementary Table 1), which were then aligned against the human transcriptome and HSV-1 strain 17 syn+ annotated reference sequence using the splice-aware aligner MiniMap2[25]. Relative transcript abundance counts for both host and viral RNAs showed high reproducibility between biological replicates (*H. sapiens* r² = 0.985, HSV-1 r² = 0.999) (Supplementary Fig. 1a), despite differing depths of sequencing, and minimal RNA decay during library construction and sequencing (Fig. 1b, c). As a final examination, we constructed an additional direct RNA-seq library from the same source material (technical replicate) and ran this on a separate MinION device, confirming that the sequencing data were also reproducible across instruments (Supplementary Fig. 1b). Satisfied that direct RNA-seq is highly reproducible, we subsequently sequenced two additional samples to enable comparisons between early (6 h) and late (18 h) time points of HSV-1 Patton infection of NHDFs, and to examine the contribution of the virion host shut-off (vhs) protein (Fig. 1a, Supplementary Table 1). Both of these yielded similar-sized datasets (Fig. 1a) with the major difference being a significantly reduced fraction of HSV-1 sequence reads in the Δvhs dataset (Fig. 1a, Supplementary Table 1), likely reflecting the involvement of vhs in host shutoff and antagonism by innate defenses at the beginning of the infection cycle[26].

**Comparing nanopore direct RNA-Seq and Illumina RNA-Seq.** We next sought to directly compare the nanopore and Illumina approaches by sequencing the polyadenylated fraction of the HSV-1 transcriptome using the same starting material. Here, profiling the genome-wide depths of coverage resulted in a similar visual profile (Fig. 2a), in which peaks corresponded to previously annotated transcription units. Using a 100-nt sliding-window approach (Fig. 2a, b), we examined changes in mean read depth (MRD) between genic and intergenic regions and determined that MRDs in genic regions were ~7–12-fold higher than in intergenic regions in both normalized Illumina (6.8-fold) and nanopore (12.1-fold) datasets (Fig. 2c). This difference presumably reflects mis-priming from internal adenosine-rich tracts during the poly(A) selection step included in standard Illumina protocols, combined with intrinsic biases in reverse transcription and amplification steps[13,27]. By contrast, the requirement for ligation of an adaptor-coupled motor protein to the poly(T) adaptor in direct RNA-seq (Fig. 1c) means that only polyadenylated RNAs will pass through the pore complex.

To obtain relative gene expression counts, mapped sequence reads are generally assigned to specific transcription units. However, the compact, gene-dense nature of the HSV-1 genome presents challenges when applying short-read Illumina sequencing strategies. Significantly, the viral genome contains multiple complex gene arrays in which distinct overlapping transcripts share the same poly(A) signal and RNA cleavage sites. This is exemplified in Fig. 2d and e in which HSV-1 gene expression counts have been generated from both nanopore and Illumina datasets and show marked differences, regardless of whether polycistronic units are treated as single "transcription units" (Fig. 2d) or are separated into their respective ORFs (Fig. 2e). While we could not extend our analysis to human transcripts due to the limited depth of sequencing, a prior study showed very high levels of correlation between nanopore and Illumina gene expression counts[22]. This reflects the less compact organization

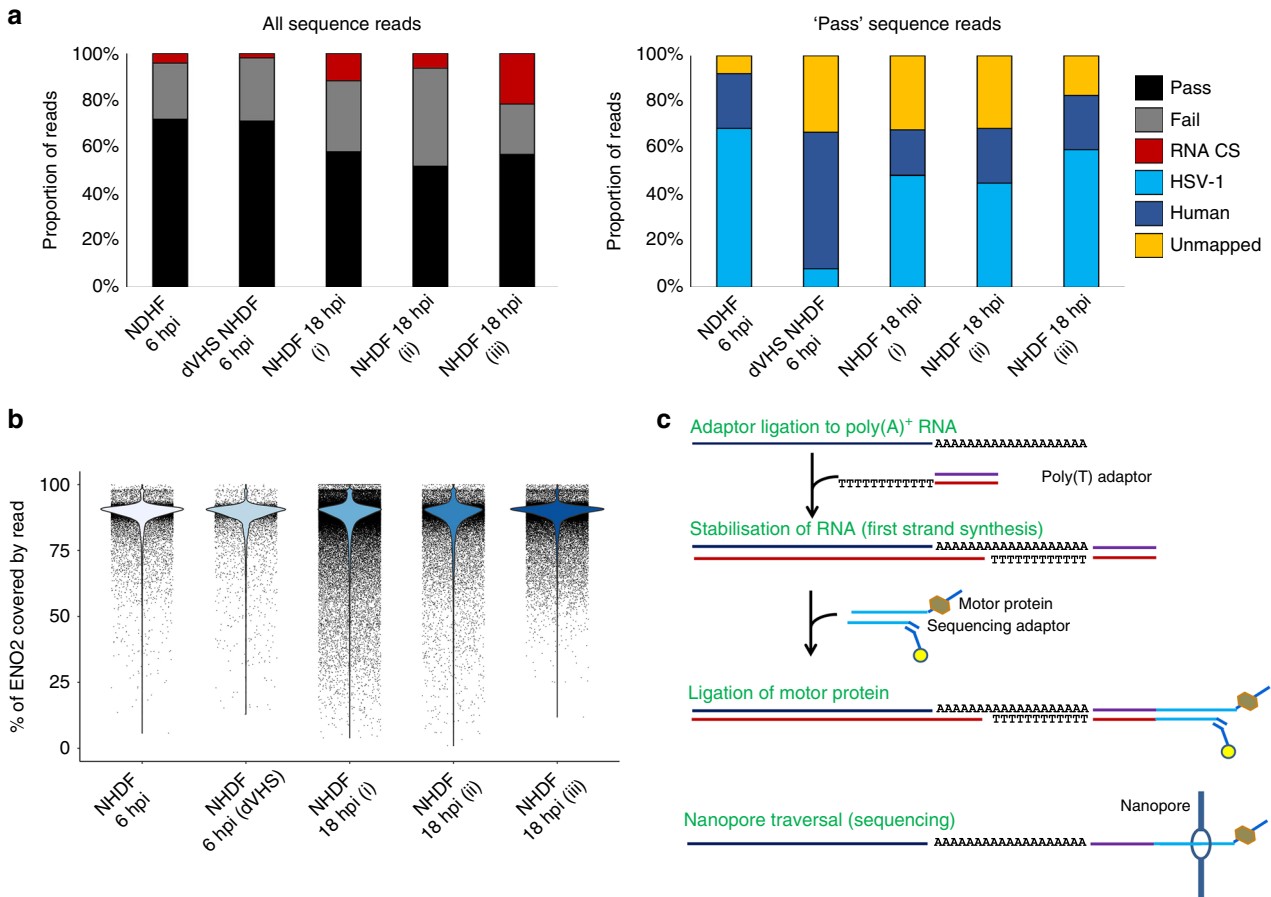

**Fig. 1** Direct RNA sequencing using nanopore arrays is highly reproducible. **a** Summary metrics for five separate direct RNA-seq runs using normal human dermal fibroblasts (NHDFs) infected with HSV-1 strain Patton GFP-Us11 or HSV-1 strain F vhs null (Δvhs) for either 6 or 18 h. NHDF 18hpi (i) and (ii) represent biological replicates, with an additional technical replicate, NHDF 18hpi (iii), performed on a separate minION device. Calibration strand reads originate from the spiked human enolase 2 (ENO2) mRNA. Pass and fail reads were classified as such by the albacore basecaller. Only reads passing QC ("pass") were retained for downstream analyses and these were classified by mapping against the HSV-1 genome and *H. sapiens* transcriptome. Only a small proportion of reads could not be mapped. **b** The spiking of ENO2 mRNA allows assessment of RNA degradation during library preparation. Here, mRNA degradation is represented by the fraction of ENO2 covered by individual reads and indicates only minimal 5′ degradation during library preparation. **c** Overview of the nanopore RNA-sequencing methodology. A poly(T) adapter is ligated to poly(A) tails and used to prime first-strand synthesis of cDNA which stabilizes the RNA strand. The poly(T) adapter also allows ligation of the motor protein required to guide the RNA strand through a nanopore

and significantly improved annotation of genes in the human genome.

**The utility of error correction and pseudotranscripts**. Each nanopore-derived sequence read represents a single full length or 5′ truncated transcript, read in the 3′ – > 5′ direction, present in the poly(A)+ RNA pool. Each read maps with high specificity but comparatively low identity (80–90% (nanopore) vs. > 99.9% (Illumina)) to the reference genome/transcriptome. One goal of our study was to utilize these data to identify novel transcript isoforms as evidence of previously unknown gene products. While this is comparatively simple in organisms where genes are for the most part arrayed as single units, the highly compact nature of viral genomes often results in multiple gene units arrayed in an overlapping manner that greatly complicates the use of short-read Illumina sequencing for such studies. As each nanopore sequence read represents a single polyadenylated RNA, it should be possible, given sufficient sequencing depth, to identify the full spectrum of transcribed, polyadenylated gene products, irrespective of overlap. Reasoning that most polyadenylated transcripts are likely translated within the infected cell, we asked if we could identify known or novel open-reading frames (ORFs) within our sequence reads and thereby determine the breadth of protein variants expressed

by HSV-1 in these cells. Unfortunately, such an analysis is hindered by the presence of numerous insertion/deletion (indel) and substitution-type errors within the raw nanopore reads, a consistent issue with nanopore sequencing[28]. To overcome this, we designed a novel error-correction strategy utilizing proovread[29], to reduce the errors present in the raw sequence read data (Fig. 3a), combined with a decision matrix operating across a range of error-corrected subsampled datasets (Fig. 3b). Briefly, we assessed the amount of error in a given read by comparing the CIGAR string length for a given read in the uncorrected dataset against corrected datasets generated using subsampled Illumina RNA-Seq data from the same source material (Fig. 3a, see the section Methods for extended description of methodology). Surprisingly, we observed that increasing the size of subsampled Illumina datasets did not always result in shorter CIGAR string lengths (our proxy for mapping accuracy). We attributed this to proovread correction performing optimally at read depths of 30–50x[29], combined with highly variable transcript abundances in both direct RNA and Illumina RNA-Seq datasets.

Error correction rescued up to 9% of unmapped raw reads (Supplementary Fig. 2a), the majority of them spliced, increasing the final yield of mapped reads. We next examined the changes in the overall read length and the alignment length. Error

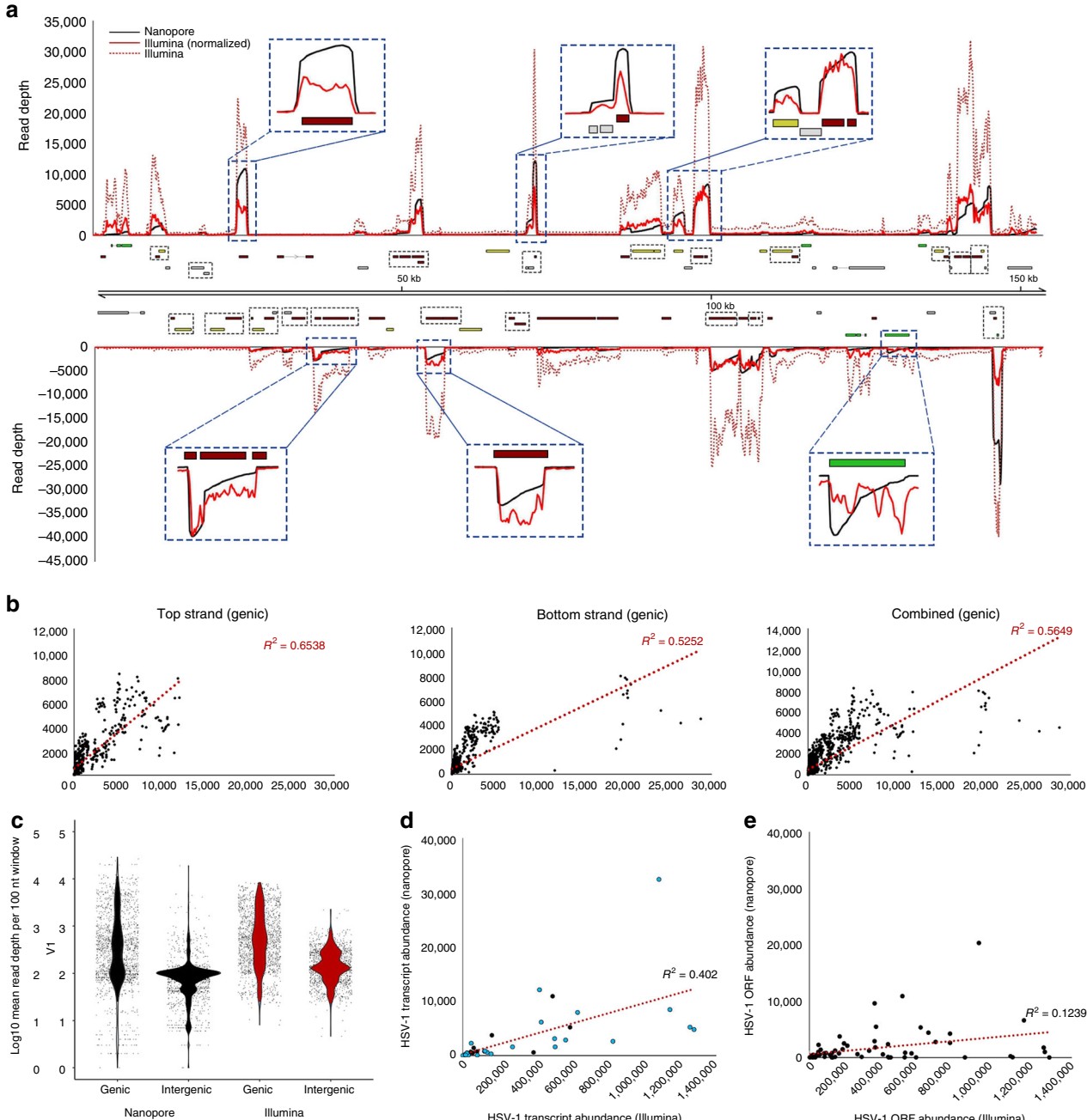

**Fig. 2** Comparison of direct RNA nanopore sequencing to Illumina sequencing. **a** HSV-1 genome-wide sliding-window (100 nt) coverage plots of poly(A) RNA sequenced by nanopore (black) and Illumina (red) technologies. Nanopore reads represent a single polyadenylated RNA, directly sequenced, while Illumina reads are derived from highly fragmented poly(A)-selected RNAs. Illumina data (red dotted line) were normalized (red solid line) to produce the same overall coverage as the nanopore data. The HSV-1 genome is annotated with all canonical open-reading frames (ORFs) and colored according to kinetic class (green—immediate early, yellow—early, red—late, and gray—undefined). Multiple ORFs are grouped in polycistronic units and these are indicated by black hatched boxes. The y-axis represents absolute read-depth counts. Inset windows (blue hatched boxes) exemplify the 3′ bias inherent to direct RNA-seq (due to sequence reads being generated 3′ −> 5′) that is less prevalent in Illumina data. **b** Correlation analyses of HSV-1 genome coverage were generated using nanopore and Illumina sequence data. The sliding-window analysis was determined by calculating and plotting mean read-depth values per 100 nucleotide windows across canonically defined genic regions in both a strand-specific and strand-combined manner. **c** Dot plots denoting read- depth values (100-nt intervals) in genic and intergenic regions for both direct RNA-seq and normalized Illumina datasets. Read depths between genic and intergenic regions differ by a mean fold difference of 12.08 (nanopore) and 6.82 (Illumina). The y-axis is log-10 scaled. **d**, **e** Transcript abundances were counted for nanopore and Illumina datasets by aligning against two versions of the HSV-1 transcriptome. The simplified version (left) collapses polycistronic units into simple transcription units, while the standard version (right) comprises all individual coding units, whether mono- or polycistronic. The impact on comparative transcript abundance estimates is greater in the latter

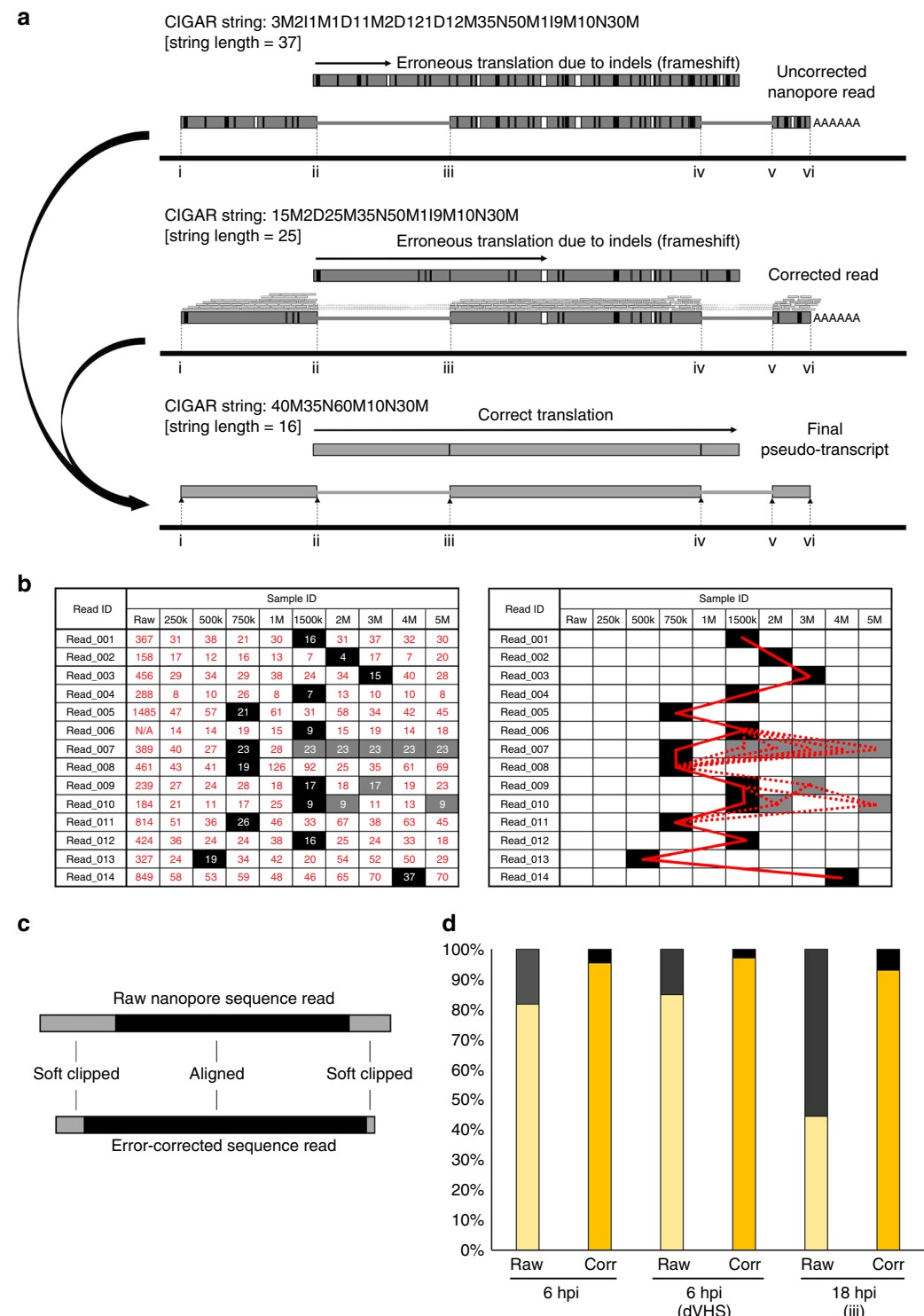

correction marginally reduced the overall read length while increasing the alignment length (Fig. 3c, d, Supplementary Fig. 2b). Our approach showed that (i) error correction notably reduces the numbers of indel and substitution-type errors in all nanopore reads, (ii) the impact of Illumina read subsampling was minimally affected by the size of the subsampled dataset, and (iii) error correction rescued reads that could not previously be mapped and increased alignment length within individual reads.

Despite the relative success of this error-correction approach, sufficient numbers of indel-type errors remained to preclude accurate identification of encoded ORFs in over 80% of

transcripts (Supplementary Figs. 3 and 4). We therefore used the mapping positions resulting from alignment of the error-corrected nanopore reads to generate "pseudo-transcripts". Each pseudo-transcript contains the same alignment start, stop, and internal splicing sites as the error-corrected read, but with the transcript sequences corrected by substituting in the corresponding reference genome sequence (Fig. 3a), thus removing any remaining indels and mutations that would otherwise generate a nonsense frameshift. We subsequently compared the pseudotranscripts generated with or without error correction by determining the proportion of sequence reads encoding ORFs longer than 90

**Fig. 3** Error correction and generation of pseudotranscripts to overcome sequencing errors inherent to the nanopore method. **a** Raw nanopore reads include numerous indel and substitution errors that hinder the identification of encoded ORFs and thereby impede annotation of the transcriptome. Illumina datasets generated from the same material allowed error correction using proovread (and see Figure S2). Subsequently, the transcript start/stop positions and internal splice positions were used to generate pseudotranscripts free of indel and substitution errors that permit unambiguous ORF prediction. Example changes in CIGAR string lengths for a given read are shown for each step of correction. **b** To optimize proovread error correction, we tested a range of subsampled Illumina datasets and evaluated corrected reads by the length of the CIGAR string (see Methods). Because optimal Illumina subsampling varies between reads, we subsequently applied a decision matrix utilizing the best-corrected version of a given read (filled boxes) as scored by the shortest CIGAR string length. Where multiple subsampling sets produce identical shortest CIGAR scores (shaded boxes), no difference was observed between the resulting sequences. The bold red line indicates the path chosen (i.e., from which error-corrected dataset a given read was drawn), while the dotted lines indicate alternative paths that produce the exact same result due to having identical CIGAR string lengths. **c** Schematic representation of the effect of error correction. The overall length of error-corrected nanopore reads is marginally less than raw sequence reads but the aligned portion of error-corrected reads is longer. **d** For each sequence read, the longest encoded ORF (>90 nt) was identified. Here, error correction notably increases the proportion of sequence reads containing translatable ORFs. In other words, the removal of indel errors improves our ability to identify novel and known ORFs

nt (equivalent to 30 codons). Here, 44–85% of uncorrected pseudotranscripts encoded ORFs > 90 nt, compared with 93–97% for error-corrected pseudo-transcripts (Fig. 3d). This showed that using error-corrected nanopore reads as seeds to generate pseudo-transcripts improves the identification of putative protein sequences. To this end, we successfully identified full-length mRNAs encoding the expected protein products for all of the canonical HSV-1 genes except for UL36 (9420 bp) and UL52 (3177 bp), two of the longest ORFs.

**Mapping the starts and ends of viral transcripts**. Another goal of this study was to re-evaluate the coding capacity of HSV-1. The distribution of nanopore read lengths for both viral and host transcripts remained similar at discrete sampling times (6 and 18 hpi) and interestingly, were not obviously different for the HSV-1 Δvhs mutant (Supplementary Fig. 5). It is possible that the vhs endonuclease has limited impact on the viral transcriptome at the times examined, or that once cleaved, mRNAs are degraded very rapidly, consistent with the reduced read frequency for viral but not host transcripts in cells infected with the mutant. While mRNAs mapping to each canonical HSV-1 ORF were detected in at least one of the datasets, transcripts corresponding to several ORFs were detected at very low levels in both the direct RNA-seq and Illumina datasets. These include RS1 (ICP4), UL9, UL15, UL36, and the LAT precursor and their low abundance can be variously explained by nontraditional kinetics, mRNA length, or mRNA stability. For example, during productive infection, the 8.3-kb polyadenylated LAT precursor might be underrepresented because it is one of the last viral transcripts to accumulate during productive infections[10]. Similarly, ICP4 is an immediate–early gene that is reported to be expressed at comparatively low levels during infection[21].

Peaks representing the 5′ and 3′ ends of sequenced poly-adenylated RNAs map closely to previously established transcription start and termination sites of several HSV-1 transcription units (Fig. 4), and we term the peak locations as either proximal transcription start sites (pTSS) or proximal transcription termination sites (pTTS) to reflect the fact that these are approximate rather than exact indicators of the actual capped 5′ end or the post-transcriptionally processed 3′ end. We noted canonical TATAAA boxes upstream of 13 HSV-1 genes, with the maximal pTSS peak positioned 30–48 bp downstream. We cataloged proximal pTSS sites for all canonical HSV-1 ORFs except for UL9, UL36, UL52, and RS1, as well as the latency-associated transcript precursor (Supplementary Table 2). While the use of these pTSS was consistent between the 6- and 18-h time points, several ORFs (UL6, UL8, UL14, UL24, UL29, UL51, UL53, and US1) were notable for having multiple pTSS (Fig. 4, Supplementary Figs. 6 and 7, Supplementary Table 3), resulting in either

elongated 5′ UTRs or 5′ truncated transcripts. Moreover, we observed eight pTSS located within previously defined ORFs (UL6, UL12, UL24, UL30, UL41, UL44, UL53, and UL54; Supplementary Table 4), potentially encoding novel (different reading frame) or alternative (same reading frame) protein products or conceivably, polyadenylated noncoding RNAs (ncRNAs). By contrast, cataloging the pTTS sites revealed that sites of cleavage and polyadenylation were 6–103 bases (median 22) downstream of canonical AAUAAA poly(A) signal sites (PAS, Supplementary Figs. 6 and 7). Importantly, the accuracy of our pTTS estimates was demonstrated by their positioning within 0–3 bases (median 1) of experimentally determined 3′ ends of transcripts covering all of the unique short-region ORFs except US2[30].

**Read-through transcription increases HSV-1 coding capacity**. Read-through transcription or the traversal of a PAS without downstream RNA cleavage, polyadenylation, and detachment of the RNA pol II machinery[31], is induced in the host by HSV-1 infection of human foreskin fibroblasts[19] and has been posited as another mechanism by which the infecting virus disrupts host gene expression. While that study found no evidence of read-through transcription occurring in the viral genome, visual inspection of HSV-1 sequence read alignments generated from direct RNA-seq data provided evidence of read-through transcription at both 6 and 18 h (Supplementary Fig. 8).

To examine read-through transcription in detail, we segregated sequence reads according to the number of AAUAAA motifs present in each one and re-aligned these against the HSV-1 genome (Fig. 5a). We observed that 65–69% of viral polyadenylated RNAs contained a single consensus AAUAAA PAS that was assumed to signal for 3′-end processing. A further 7–15% contained two AAUAAA motifs, while up to 3% contained three or more AAUAAA motifs. Overall, 20–25% of sequence reads contained no AAUAAA motif (Fig. 5a, b). The read coverage profiles generated for each group of sequence reads were generally similar except in specific loci (Fig. 5a). One such exception is exemplified by RL2 (ICP0) mapping reads, which contain two neighboring AAUAAA motifs (inset panel (i) in Fig. 5a). We determined that pTTS sites for reads with no AAUAAA motifs mapped upstream of AAUAAA motifs and resulted from 3′ truncation during either error-correction or alignment (soft-clipping). For all other sequence reads, pTTS sites mapped 3′ to the AAUAAA motifs (inset panel (ii) in Fig. 5a). Thus, all HSV-1 genes, whether organized individually or in complex gene clusters, appeared to utilize canonical AAUAAA motifs to signal transcription termination with only four exceptions. For the US8/US8A/US9 gene cluster (inset panel (iii) in Fig. 5a), the UL52/UL53 gene cluster, the UL4/UL5 gene cluster, and RL1, a noncanonical AUUAAA motif is utilized instead. We noted

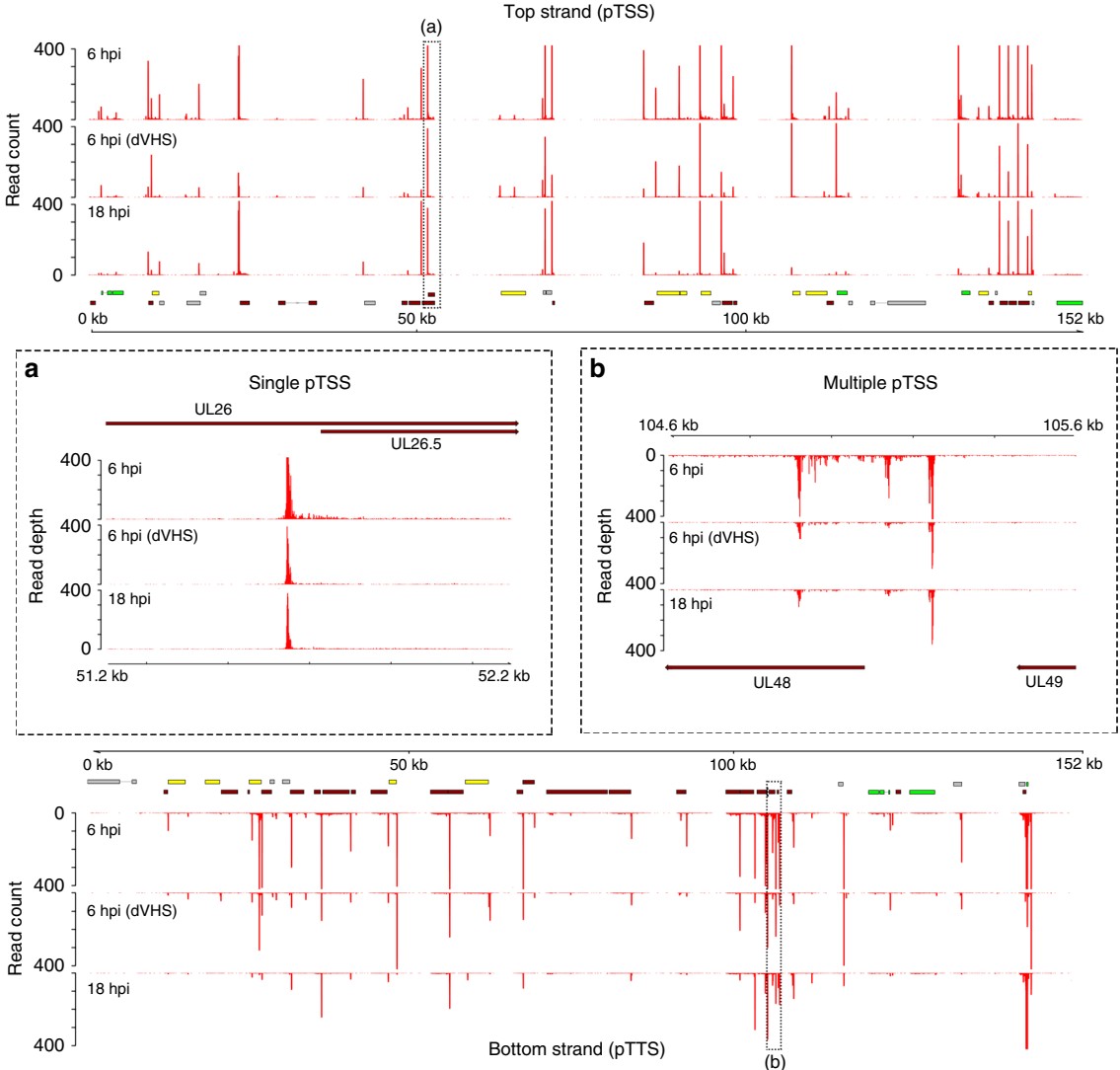

**Fig. 4** Viral polyadenylated RNAs initiate at single or multiple locations. To visualize transcription start sites, the extreme 5′ end of each nanopore read was plotted against the HSV-1 genome. Datasets correspond to NHDF infected with HSV-1 strain Patton for 6 hpi (upper track), strain F Δvhs for 6 hpi (middle track), and strain Patton for 18 hpi (lower track). Peaks corresponding to clustered 5′ ends, are referred to as proximal transcription start sites (pTSS) and likely differ by only a few nucleotides from the actual capped 5′ end. In 13 cases, the pTSS is positioned 30–48 bp downstream of a canonical TATA box. Upper panel: top strand. Lower panel: bottom strand. Inset boxes: **a** Transcription of the HSV-1 UL26.5 gene initiates at a single location throughout infection. **b** UL48 transcription initiates at multiple locations, one of which is internal to the canonical UL48 ORF, suggesting transcripts encoding a truncated or alternative protein. Canonical HSV-1 ORFs are colored according to kinetic class (IE—green, E—yellow, L—red, and undefined—gray), while polycistronic transcriptional units are indicated by hatched boxes

that the major peaks in the 3+ AAUAAA dataset generally corresponded to the presence of HSV-1 gene clusters and were consistent with read-through. Finally, the relative proportions of transcripts containing differing numbers of AAUAAA motifs did not significantly change between 6 and 18 hpi (Fig. 5b), indicating that read-through transcription is not a specific feature of early or late infection times.

While read-through transcription across the host genome generally leads to aberrant non-adenylated transcripts[19], here, the compact nature of the HSV-1 genome enables RNA pol II transcription to terminate following traversal of poly(A) signal sites further downstream, often at the end of adjacent single or polycistronic transcription units (Fig. 5c, Supplementary Figs. 8 and 9). We considered that read-through transcription could provide a simple mechanism for generation of chimeric poly-adenylated RNAs. Chimeric RNAs in mammalian cells are thought to arise predominantly from *trans*-splicing[32] but in compact,

gene-dense viral genomes, read-through could produce multi-ORF transcripts in which *cis*-splicing[33] between neighboring ORFs creates novel fusion proteins (Fig. 5c). To search for examples, we predicted the translation products of all spliced HSV-1 transcripts and compared the results to a database of all canonical translated ORFs. This revealed two distinct fusions of neighboring ORFs (RL2–UL1 and UL52–UL54), (Figs. 6 and 7, Supplementary Table 5) which we predict result from read-through transcription.

**A chimeric mRNA encodes an ICP0 and glycoprotein L fusion.** The usage of both the UL52–UL54 (Fig. 6a) and RL2–UL1 (Fig. 7a) splice junctions can be readily detected by end-point RT-PCR, albeit at lower levels than their non-fusion forms, using RNA collected from NHDFs infected by multiple HSV-1 strains at 18 h post infection (Figs. 6b and 7b). Conversely, neither the RL2–UL1 nor UL52–UL54 splice is detected if protein synthesis is blocked using cycloheximide (CHX) or if viral DNA replication is

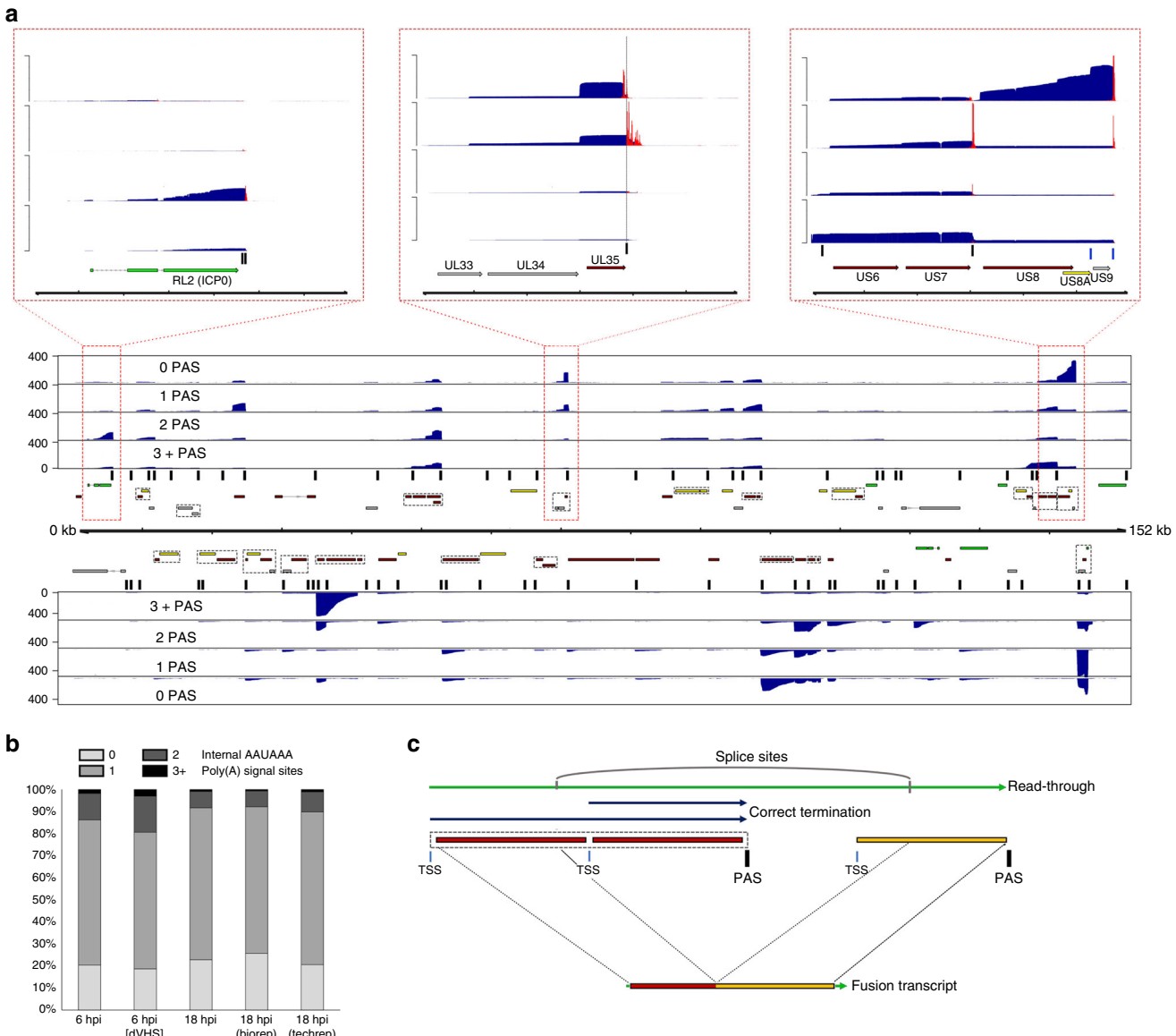

**Fig. 5** Detection of read-through transcription from the HSV-1 genome. **a** HSV-1 sequence reads were segregated according to the number of AAUAAA PAS motifs present and aligned against the HSV-1 genome to produce coverage plots showing the location of mapped reads. Here, complex (multiple overlapping ORFs) gene arrays are identified by cross-hatched boxes, while the locations of AAUAAA motifs are indicated by vertical black bars. The three inset boxes correspond to the red cross-hatched areas of the genome that exemplify (left) the presence of two AAUAAA motifs at the 3′ end of the RL2 transcript, (center) the position of pTTS sites relative to AAUAAA motifs (black vertical line), and (right) usage of the non-canonical AUUAAA motif (blue vertical line). pTTS estimates are shown as red overlays on the inset coverage plots. **b** HSV-1 transcription termination is generally initiated by recognition of a canonical (AAUAAA—dark gray) PAS sequence. Evidence of read-through transcription includes the presence of multiple AAUAAA motifs within a transcript and is observed in a small proportion 1–3% of HSV-1 mapping direct RNA-seq reads. **c** Transcription of HSV-1 genes initiates at transcription start sites (blue vertical line) and typically terminates shortly after traversing a canonical (AAUAAA) PAS sites (black vertical line). In rarer cases, termination does not occur and transcription extends further downstream as read-through until another PAS site is used. These extended transcripts may be subject to internal splicing which can give rise to fusion ORFs

blocked using phosphonoacetic acid (PAA) or if cells are infected with an ICP4-null mutant defective for early and late gene expression as well as viral origin-dependent replication (Figs. 6b and 7b). This indicates that expression of both RL2–UL1 and UL52–54 requires viral DNA replication. The late kinetics are also evident when the abundance of the UL52–UL54 and RL2–UL1 splices are compared with the canonical UL52, UL54, RL2, and UL1 genes by real-time RT-qPCR at different times post-infection (Figs. 6b and 7b). This is notable because RL2 and UL54 are considered immediate–early genes, while UL52 is an early gene and UL1 a late gene.

Although canonical UL1 transcripts are not thought to be spliced, the internal UL1 splice acceptor motif GTAG|G is perfectly conserved across all of the HSV-1 full or partial genome sequences available in GenBank ($n = 155$), but is not present in HSV-2 genome sequences (Supplementary Table S5). Similarly, the splice donor and acceptor sequences utilized in the UL52–UL54 fusion are likewise perfectly conserved (Supplementary Table S5). Production of these new transcripts is not unique to NHDFs because both the RL2–UL1 and UL52–UL54 splice junctions could also be detected by RT-PCR in HSV-1-infected human ARPE-19 retinal pigment epithelial cells and hESC-derived neurons (Figs. 6b and 7b).

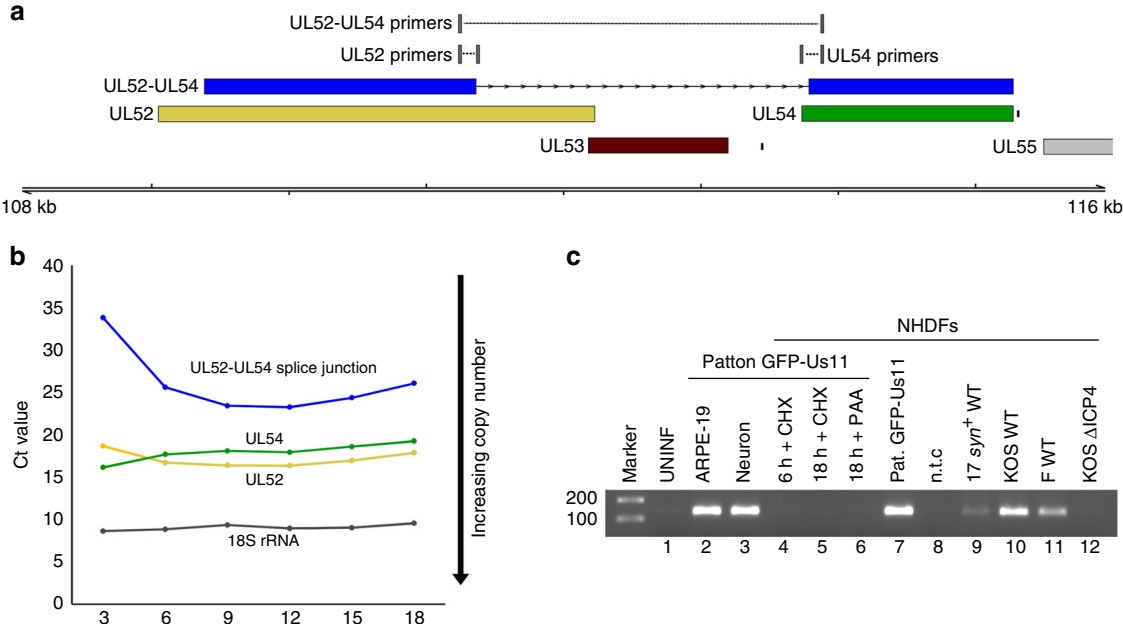

**Fig. 6** Chimeric UL52–UL54 mRNA is expressed with late kinetics and by multiple HSV-1 strains. **a** The UL52–UL54 fusion transcript. **b** Assessment of the UL52–UL54 splice junction usage at different times after infection by real-time RT-qPCR. Increased RNA abundance is reflected as lower crossover threshold (Ct) values and normalized to 18S rRNA. Three technical replicates were utilized per condition/time point. Representative data are shown from one of three biological replicates. **c** Detection of the unique UL52–UL54 splice junction by RT-PCR using a primer scanning the splice junction. NHDFs were infected in parallel with either HSV-1 strain Patton (lanes 2–7) or with wild-type strain 17 syn+ (lane 9), KOS (lane 10), strain F (lane 11) viruses, or with n12, a KOS ICP4 null mutant (lane 12), and RNA was collected at either 6 h (lane 4) or 18 h (lanes 2–3, 5–7, and 8–12) post infection. Inhibitors of protein synthesis (cycloheximide, CHX) or the viral DNA polymerase (phosphonoacetic acid, PAA) were included as indicated (lanes 4–6). Amplification products were visualized with ethidium bromide

The in-frame fusion of ICP0 (ORF RL2) and glycoprotein L [gL] (ORF UL1), arises through splicing from the canonical exon 2 splice donor within the coding sequence of RL2 to a previously unknown splice acceptor with the UL1 ORF, such that the first two exons of RL2 (ICP0 residues 1–241, including the RING finger domain) are fused in-frame to the last 191 bp of the UL1 ORF, corresponding to the residues 162–224 of gL (Fig. 7a). The ICP0-gL fusion protein has a predicted molecular mass of 32 kDa and a band of this size was detected at late (18 or 24 h) but not early (6 h) times in HSV-1 strain Patton-infected cells by immunoblotting using an antibody that recognizes the gL C terminus (Fig. 7c lanes 3–6). To verify this band as the fusion protein rather than a gL glycosylation intermediate, immunoprecipitations were performed using either a monoclonal antibody that recognized the ICP0 N terminus (lanes 2 and 8) or an isotype-matched antibody as a specificity control (1 or 7). Immunoblotting of the recovered material with either anti-ICP0 or anti-gL detected a diffuse band of similar size in the ICP0-specific immunoprecipitation (lanes 2 and 8) that was absent in the control (lanes 1 and 7). This demonstrates that a 32-kDa protein is detected in HSV-1-infected cells at late times post infection that has antigenic determinants derived from the ICP0 N-terminus and the gL C-terminus. Moreover, it confirms that a novel HSV-1 mRNA first identified using direct RNA sequencing encodes a polypeptide produced in virus-infected cells.

## Discussion
Decoding the transcriptional landscape of viral pathogens is a vital first step in understanding how they overcome host cell defense mechanisms and ultimately, gain control of the host transcriptional and translational machinery. The compact, gene-dense nature of dsDNA viruses poses a special challenge when using conventional RNA-Seq strategies for such an analysis because of the reliance on short fragments of recoded RNA (Illumina sequencing)[27]. Recent studies have illustrated the potential of herpesviruses to encode a much broader range of transcript isoforms than previously thought, thereby expanding the potential proteome of these viruses and opening up the possibility of new viral functionalities or more nuanced interactions with the host cell[6,18].

To address this, we have systematically examined the utility of direct poly(A) RNA sequencing using nanopore arrays to profile a highly complex large dsDNA viral transcriptome. We have demonstrated the efficacy and reproducibility of this method and shown that the intrinsic problems associated with the high error rate of nanopore sequencing can be partially overcome by error-correcting using short-read (Illumina) sequencing data and the generation of pseudotranscripts. This approach improves read alignment and identification of coding sequences within individual transcripts. Nanopore sequencing of native poly(A) RNA species is particularly effective when applied to the initial characterization of pathogen transcriptomes, because it allows investigators to rapidly define transcript structures and/or splicing profiles, distinguish mono- and polycistronic transcription units, and provide comprehensive maps of transcription initiation sites and transcript termini. However, while Illumina-based error correction and the generation of pseudotranscripts significantly improves our data, it is unlikely to be suitable for applications beyond the analysis of transcript isoforms. Specifically, the advantage of pseudotranscripts is that they greatly improve our ability to identify coding sequences (ORFs) within individual transcripts. However, generating them also obscures potentially informative SNP and indel mutations and requires users to carefully balance their analyses by incorporating raw and error-corrected nanopore reads, as well as inferred pseudotranscripts to

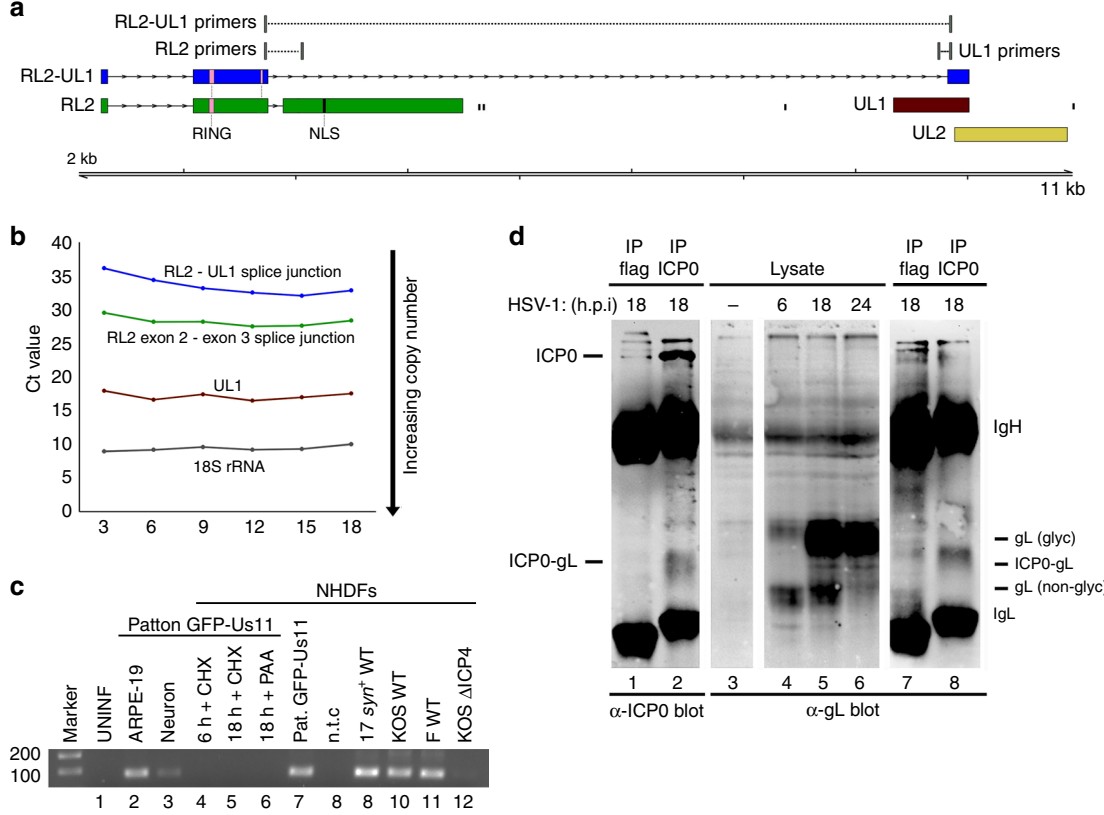

**Fig. 7** Chimeric RL2–UL1 mRNA is expressed with late kinetics and by multiple HSV-1 strains. **a** The RL2–UL1 fusion transcript encodes an ICP0-gL fusion protein that lacks two phosphorylation sites (P2, P3) and the nuclear localization signal (NLS) domain present in ICP0. **b** Assessment of canonical RL2 exon2–exon3 and novel RL2 exon 2—UL1 internal splice junction usage at different times after infection by real-time RT-qPCR. Increased RNA abundance is reflected as lower crossover threshold (Ct) values and normalized to 18S rRNA. Three technical replicates were utilized per condition/time point. Representative data are shown from one of three biological replicates. **c** Detection of the unique RL2 exon 2—UL1 splice by RT-PCR using a primer scanning the splice junction. NHDFs were infected in parallel with either HSV-1 strain Patton (lanes 2–7) or with wild-type strain 17 *syn+* (lane 9), KOS (lane 10), strain F (lane 11) viruses, or with n12, a KOS ICP4 null mutant (lane 12), and RNA was collected at either 6 h (lane 4) or 18 h (lanes 2–3, 5–7, and 8–12) post infection. Inhibitors of protein synthesis (cycloheximide, CHX) or the viral DNA polymerase (phosphonoacetic acid, PAA) were included as indicated (lanes 4–6). Amplification products were visualized with ethidium bromide. **d** Detection of the predicted ICP0-gL fusion protein. Lysates were prepared from mock (lane 3) or HSV-1 strain Patton-infected NHDFs collected at 6, 18, or 24 h post infection (lanes 1–2, 4–8) and analyzed by immunoblotting with (lanes 1–2 and 7–8) or without (lanes 3–6) prior immunoprecipitation using anti-ICP0 (lane 2 and 8) or control anti-flag- (lanes 1 and 7) loaded protein A beads. After fractionation by SDS-PAGE, membranes were probed using primary antibodies recognizing either the N terminus of ICP0 (lanes 1–2) or the C terminus of glycoprotein L (lanes 3–8). The ICP0-gL fusion peptide has a predicted mass of 32 kDa. Additional reactive species corresponding to ICP0, glycosylated and non-glycosylated gL, and antibody heavy (IgH) or light chains (IgL) are indicated. The image shown is representative of three independent replicates

maximize accuracy. At this point, we do not recommend our approach for profiling small, rapidly evolving RNA viruses that exist as quasi-species as these are better examined using alternative approaches[34].

Even though the HSV-1 genome has been studied for decades and was a leading paradigm in studies of promoter organization, the more detailed transcriptomic data generated in this study expose a more complex spatial and temporal patterns of transcription start site usage than was previously known. Furthermore, it has also uncovered an otherwise cryptic class of viral fusion transcripts that encode novel proteins. Transcription initiation sites (TSS) are critical for the control of productive cycle gene expression as their location relative to the translation initiation site determines the length and composition of the 5′ UTR of mRNAs, which can have profound effects on translation efficiency. Although the majority of RNA polymerase II (RNAP) transcribed genes recruit TFIID to the core promoter, only a minority (~10%) contain an identifiable TATA box[35]. Our data indicate a similar pattern in the HSV-1 genome and in many cases, the actual position of the TSS may be determined through

interactions of the general transcription machinery with the viral transcription factor ICP4[36] and other core promoter elements[37]. While direct RNA-seq does not currently allow mapping of TSS at nucleotide resolution, we nevertheless observed that in many transcripts, the 5′ end of HSV-1 sequence reads mapped to between 30 and 48 bp downstream of a consensus TATA box. This level of resolution is sufficient to identify HSV-1 genes that use multiple TSS, including eight with internalized TSS that may encode truncated protein isoforms (Supplementary Table 4), four of which have been identified previously through a limited alternative long-read sequencing method[38]. While beyond the scope of this study, integrating our transcriptomic data with alternative TSS mapping approaches, such as CAGE-Seq or RNA pol II ChIP-Seq will advance our understanding of promoter usage in HSV-1 and the relationship of start site selection to subsequent steps in RNA maturation.

Inter-ORF transcription (pervasive transcription) has been observed in several γ-herpesviruses and shown to be functionally important[5]. With the exception of polycistronic mRNAs, few examples have been described to date for α-herpesviruses.

Detection of HSV-1 fusion transcripts that encode chimeric proteins was therefore exciting and unexpected. Given that the two predominant fusions we identified occur between discrete neighboring transcription units, the likely mechanism involves suppression of transcription termination and 3′-end processing leading to elongated transcripts that contain functional splice donor and acceptor sequences. This was evidenced by the presence of a subset sequence reads containing multiple PAS motifs. Intriguingly, while HSV-1 almost universally requires AAUAAA motifs to signal transcription termination, the relative ability of a given motif to signal termination appears dependent on genomic context. In other words, read-through transcription (or disruption of transcription termination) was most evident for poly(A) signal motifs located within complex gene arrays, potentially suggesting a role for sequence constraints in flanking sequence regions. Our findings contrast with an earlier Illumina-based sequencing study of nascent transcripts in HSV-1-infected fibroblasts that reported an absence of read-through transcription of viral genes during the first 8 h of infection[19]. This likely reflects the differences in the sequencing approaches and analyses methods used in the two studies. Rutkowski et al. sequenced rRNA-depleted total RNA collected by Illumina sequencing, an approach that allowed a comprehensive examination of read-through in both host and viral genomes during the first 8 h of infection. A re-analysis of that dataset indicates that read-through transcription of viral genes does occur at low levels by 8 hpi (Supplementary Figs. 8 and 9) but may have been missed, or dismissed as background, due to its low level in the original study. By contrast to the approach utilized by Rutkowski et al.[19], direct RNA sequencing only sequences mature polyadenylated RNAs at later infection times (6–18 hpi), allowing us to examine this phenomenon from an alternative viewpoint and show that the read-through transcription of the viral genome becomes more prevalent at late infection time points. Unlike host transcripts for which neighboring PAS are located far apart (tens to hundreds of kilobases), viral read-through transcripts are still able to efficiently terminate because the more compact genome provides numerous alternative downstream poly(A) signal sites within a few kilobases. Moreover, the compact nature of the viral genome limits the window in which read-through transcription can be observed by short-read sequencing approaches. While read-through transcription remains the most plausible explanation for the generation of fusion transcripts, we cannot yet entirely exclude the possibility that they instead arise from *trans*-splicing events that join separate mRNAs. Although rare outside of protozoan parasites, there are examples of intergenic splices between viral mRNAs from JC virus and SV40, as well as rarer hybrids between viral and host transcripts[32,39,40].

To better understand whether read-through transcription in HSV-1 may produce functional proteins, we focused our subsequent analyses on the predicted ICP0-gL fusion protein encoded by the RL2–UL1 chimeric transcript. While further characterization is required, sensitivity to a viral DNA replication inhibitor and accumulation of the RNA at later times in the infection imply a regulated pattern of expression. Similarly, the conservation of the otherwise unused UL1 splice acceptor sequence among all sequenced wild-type HSV-1 genomes, combined with detection of the RL2–UL1 splice in NHDFs infected by four different HSV-1 strains, argues for a functional role for this previously unknown protein. A broader question is whether these viral chimeric mRNAs arise simply as a consequence of virus-induced transcription read-through targeting the host as part of its host shut-off strategy or are themselves a reason for the virus to interfere with termination mechanisms.

Technological advances continually redefine our abilities to ask complex questions of biological systems, such as the interplay between host and viral transcriptomes during cellular infections. This study illustrates the value of applying direct RNA sequencing to complex viruses and provides a roadmap for researchers interested in examining host–virus interactions that may eventually be extended to other pathogens (i.e., bacteria, parasites) with transcriptionally complex genomes. The biological insights presented here raise the intriguing question of whether other dsDNA viruses deliberately manipulate the host transcription and RNA-processing machinery to increase the diversity of their own proteome. Identifying specific mechanisms and determining the role of these RNA diversification strategies in viral infections will greatly enhance our understanding of host–virus interactions.

## Methods

**Cell culture, viral strains, and infection procedures**. Cells used in this study included normal human dermal fibroblasts (NHDF), human retinal pigment cells (ARPE-19 [ATCC® CRL-2302™]), and hESC-derived neurons[41]. NHDFs were cultured in DMEM supplemented with 5% FBS, ARPE-19s in DMEM/F12 supplemented with 10% FBS, and neuronal cultures as previously described[41]. For HSV-1 infections, we utilized multiple viruses corresponding to four different HSV-1 strains, including GFP-Us11 Patton[24,41], Kos[42], 17syn+, F[43], KOS N12 (ΔICP4)[44], and strain F R2621 (Δvhs)[45], always at a multiplicity of infection (MOI) of three for either 6 or 18 h prior to collecting total RNA or protein. Note that FBS concentrations were halved during the infection and post-infection periods. HSV-1 GFP-Us11 strain Patton is an effectively wild-type virus that expresses a fluorescent fusion protein with true late kinetics, and has been used extensively in studies of acute infection, latency, and viral pathogenesis.

**RNA collection, extraction, and quality control**. HSV-1-infected cells were lysed in TRIzol reagent (Invitrogen) and extracted according to the manufacturer's instructions. RNA integrity (RIN) was assessed using an RNA 6000 nanochip (Agilent Technologies) on a Bioanalyzer 2100 (Agilent Technologies). Poly(A)+ RNA was isolated from 25 μg (HSV-1) or 55 μg (VZV) of total RNA using a Dynabeads™ mRNA Purification Kit (Invitrogen), according to the manufacturer's instructions—with the only adjustment being to use 133 μl of resuspended Dynabeads rather than 200 μl as this was deemed optimal for the quantity of total RNA.

**Nanopore sequencing and post processing**. Direct RNA-sequencing libraries were generated from the isolated poly(A) RNA, spiked with 0.25 μl of a synthetic Enolase 2 (ENO2)-derived calibration strand (a 1.3 kb synthetic poly(A) m+, Oxford Nanopore Technologies Ltd.) and sequenced on one of two MinION MkIb with R9.4 flow cells (Oxford Nanopore Technologies Ltd.) and an 18-h runtime. All protocol steps are described in Garalde et al.[22]. Following sequencing, basecalling was performed using Albacore 1.2.1 [-f FLO-MIN106 -k SQK-RNA001 -r -n 0 -o fastq, fast5]. Only reads present in the "pass" folder were used in subsequent analyses.

**Illumina sequencing and post processing**. TruSeq stranded RNA libraries were prepared from poly(A)-selected total RNA for the HSV-1-infected NHDFs (18 hpi) and ARPE-19 cells (18 hpi) by staff at the New York University Genome Technology Center. Following multiplexing with additional unrelated samples, paired-end sequencing (2 × 76) was performed using a HiSeq 4000 (Illumina). FLASh[46] (–min-overlap = 10 –max-overlap = 150) was used to merge overlapping reads prior to error correction.

**Error correction of direct RNA nanopore reads**. For error correction of nanopore sequence reads, we utilized proovread[29], an error-correction package initially designed for data correction for the PacBio long-read sequencing platform. We generated error-corrected nanopore datasets by applying proovread to nanopore sequence reads using subsampled FLASh[46] compacted Illumina RNA-Seq datasets (250,000–5,000,000 paired-end reads) generated from the same material. As a metric, we examined per-read changes in CIGAR string lengths, reasoning that correction of indels and substitution-type errors would reduce string lengths and improve splice-site usage identification (Supplementary Figs. 3 and 4). Here, the length (number of characters) of the CIGAR string is long where insertions or deletions (indels) are present. Error correction reduces the number of indels present, which in turn merges match/mismatch units, reducing the overall string length (Fig. 3a). The optimal error-corrected read was considered to have the shortest CIGAR string length among all subsampled datasets (Fig. 3b). We profiled changes in CIGAR string lengths for all HSV-1 mapping and human mapping reads in our datasets and observed similar results, especially where more than 1 million subsampled Illumina reads were used for correction (Supplementary Figs. 2 and 3). However, we also observed that for a given read, the change in CIGAR string length did not consistently improve as more subsampled Illumina reads were included. We thus implemented a decision matrix that compared CIGAR string lengths for a given read using each subsampled Illumina dataset. The corrected read with the shortest CIGAR string length was considered the

best-corrected version of the read and retained for subsequent analyses. Where multiple subsampled datasets provided the same CIGAR string lengths, we observed that all the corrected sequence reads were identical in all cases. A formal evaluation of how error correction impacted on splice junction mapping was not possible due to unexpected sequencing artifacts (detailed in Supplementary Note 1).

**Generation of pseudotranscripts**. While error correction significantly reduced the number of indel-type errors in our nanopore reads, it rarely removed all of them. Thus, attempts to identify and translate ORFs within these reads produced frameshift errors that obscured the likely ORF sequence. However, by leveraging the start, stop, and internal splicing site co-ordinates from the mapping output, we were able to generate what we term "pseudo-transcripts" by substituting in the corresponding reference genome sequence in place of the existing nanopore read sequence (Fig. 3a). This served to repair sequences to an extent that internal ORF predictions could be made. Note that while error correction is not strictly required for the generation of pseudotranscripts, the improved mapping accuracy, particularly of splice junctions, justifies the additional computational time required.

**Mapping and analyses of nanopore sequence data**. Following basecalling with Albacore, nanopore reads are separated into three folders (pass, fail, and calibration strand). We used only the reads in the pass folder for subsequent analyses, and for each dataset, we ran the analyses with both raw (uncorrected) and proovread-corrected datasets. Nanopore read data were aligned to the HSV-1 strain 17 A reference genome (NC_001806) and transcriptome, as well as the *Homo sapiens* HG19 genome and transcriptome using MiniMap2[25] (-ax splice -k14 -uf –secondary = no), a splice-aware aligner. Due to many HSV-1 genes being arrayed as polycistrons, we utilized two distinct versions of the transcriptome, one containing all encoded ORFs (used for internal splicing analysis and isoform clustering), the other containing all encoded transcriptional units (utilized for examining transcript boundaries and fusion transcript discovery). Note that following mapping, all SAM files were parsed to sorted BAM files using SAMtools v1.3.1[47]. Basic analyses of sequence reads (overall lengths, alignment lengths, internal coding sequences, etc.) are detailed in Supplementary Note 2.

**Transcript boundaries analysis**. BAM files containing read data mapped to the HSV-1 genome were parsed to BED12 files using BEDtools[48], separated by strand, and the extreme 5′-mapping and 3′-mapping sites identified for all mapping reads (see Supplementary Note 2 for coding examples). The resultant dataset is a four-column file specifying each unique 5′ start site and 3′ end site identified and the number of distinct transcripts utilizing that start site.

**Fusion transcript analysis**. BAM files containing read data mapped to the HSV-1 transcriptome (transcriptional units) were parsed to identify putative chimeras (SAM flag "SA:Z"). Each chimeric sequence read was translated to identify all ORFs >30 amino acids using ORFfinder (https://www.ncbi.nlm.nih.gov/orffinder/). The resulting peptide sequences were queried (blastp[49]) against a blast database comprising all canonical HSV-1 proteins. The resulting data were manually parsed to identify peptide sequences mapping to two or more ORFs present in distinct transcriptional units. Subsequent visualization using IGV[50] was used to identify splice acceptor and donor sequences and to verify mapping integrity (see Supplementary Note 2 for coding examples).

**Data visualization**. Figures showing read data overlaying genome schematics were generated in RStudio (http://www.rstudio.com) using GViz and GenomicFeatures packages[51,52]. Additional details are specified in Supplementary Note 2.

**Experimental validation of splice-site usages**. For analysis of the UL52–UL54 splice junction, primers were designed to discriminate between canonical UL52, canonical UL54, and usage of the UL52–UL54 splice junction (Fig. 6, Supplementary Table 6). For analysis of the RL2–UL1 splice junction, primers were designed to discriminate between usage of the canonical RL2 exon2–exon3 splice junction, the novel RL2 exon—UL1 internal splice junction, and canonical UL1 (Fig. 7, Supplementary Table 6). Primers designed against the *H. sapiens* 18srRNA were utilized as a control. cDNA was generated from 500 ng of the total RNA using qScript cDNA SuperMix XLT (Quanta Bio) and 50 ng of cDNA used per PCR/real-time quantitative PCR (qPCR) reaction. For PCR amplification, the Herculase II Fusion DNA Polymerase (Agilent Technologies) was used according to the manufacturer's instructions in a 25-µl reaction volume and with 35 amplification cycles. Ct values for viral products, where detectable, always exceeded Ct 38 in negative controls (RT and uninfected NHDFs). PCR products were visualized alongside a GeneRuler 100-bp ladder (Thermo Scientific) on a 1.8% agarose gel stained with ethidium bromide. For RT-qPCR, all reactions were carried out in triplicate using a reaction volume of 25 µl and the SsoAd-vanced™ Universal SYBR® Green Supermix (BIO-RAD). All samples were run on a CFX96-Touch (Bio-Rad).

**Immunoprecipitation and immunoblotting**. Cells were lysed in 1× cell lysis buffer (Cell Signaling), fractionated by 15% sodium dodecyl sulfate polyacrylamide gel electrophoresis (SDS-PAGE), and transferred to nitrocellulose membranes (Whatman). Membranes were blocked in 5% milk diluted in Tris Buffered Saline with Tween® 20 (TBS-T) for 1 h at room temperature and incubated overnight at 4 °C with α-ICP0 (1:200, 53070, Santa Cruz Biotechnology), or α-gL[53] (1:1000, H1A259-100, Virusys Corp.) primary antibodies and detected using a horseradish peroxidase-conjugated α-mouse secondary antibody (diluted 1:5000, Sigma-Aldrich) with incubation at room temperature for 1 h and visualization by chemiluminescent detection using SuperSignal™ West Femto Maximum Sensitivity Substrate (ThermoFisher). For immunoprecipitation, lysates were incubated for 1 h at 4 °C with α-ICP0 or α-FLAG-loaded protein G agarose beads (Cell Signaling), recovered by low-speed centrifugation, and washed in TBS before heat denaturation in sample buffer. Images were captured using an iBright FL1000 (Invitrogen).

## Data availability

Raw fast5, basecalled fastq (both nanopores), and Illumina fastq datasets generated as part of this study can be downloaded from the European Nucleotide Archive (ENA) under the following study accession: PRJEB27861. The authors declare that all other data supporting the findings of this study are available within the article and its Supplementary Information files, or are available from the authors upon request.

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

## Acknowledgements

We thank Hannah Burgess for the R2621 (Δvhs) virus stocks and other members of the Mohr and Wilson Labs, as well as Werner Ouwendijk for useful input, Gary Cohen for the kind gift of gL antibodies, and N. DeLuca (University of Pittsburgh) and B. Roizman (University of Chicago) for generously providing mutant viruses. Sequencing was initiated through the NYU Langone Medical Center Genome Technology Center (GTC), which receives support from the National Institutes of Health (NIH) through a grant from the National Center for Advancing Translational Sciences (NCATS UL1 TR00038), and a Cancer Center Support Grant (P30CA016087) to the Laura and Isaac Perlmutter Cancer Center. These studies were also supported by individual grants from NIH to I.M. (AI073898 and GM056927) and A.C.W. (AI130618). T.S. received funding from the Ministry of Education, Culture, Sports, Science, and Technology (MEXT KAKENHI JP17H05816, JP16H06429, and JP16K21723), Japan Society for the Promotion of Science (JSPS KAKENHI JP17K008858), the Takeda Science Foundation, and Daiichi Sankyo Foundation of Life Science.

## Author contributions

D.P.D., I.M., and A.C.W. designed the study. T.S. and Y.M. provided additional input into study design. D.P.D., K.P.S., and T.S. performed the experiments. D.B. and D.G.P. generated and provided cultured neurons. D.P.D. analyzed all data. D.P.D., I.M., and A.C.W. wrote the paper. All authors read and approved the final paper.

## Additional information

**Competing interests:** The authors declare no competing interests.

