## [Peer Review File · Nature Communications]

Reviewers' Comments:

Reviewer #1:

Remarks to the Author:

The study presents the generation and analysis of long read RNA-seq using nanopore arrays, in order to examine the transcriptome of the HSV1 virus following infection.

The authors also generate parallel illumina short-read mRNA-seq for the same samples, and present a novel method for improvement of mapping and splice junction assignment.

Finally, they show that readthrough transcription occurs post HSV1 infection, and that subsequent novel splicing events within readthrough RNAs lead to chimera RNAs and proteins, one of which could be detected by WB.

The findings of the authors regarding readthrough and subsequent inter-gene splicing leading to chimera proteins are very interesting. However, given the current state of analyses, they remain anecdotal. No rigorous statistics or data is presented for readthrough, inter-gene splicing, or both, not even for the single RL2-UL1 chimera. As figure 6 is very convincing, especially the WB, it is highly likely that these chimeras exist, and result from readthrough. This is a major novelty factor, however without more rigorous analyses one cannot really establish its prevalence in the data, and the biological impact. A thorough comparison to illumina RNA-seq should be made in that regards, as well as to previously published nascent RNA-seq by Rutkowski et al.

The manuscript figures are mainly genome tracks. These are good as illustrations, but should accompany figures illustrating general statistics of the data to support the authors claims. This is true with regards to splicing rates, general readthrough rates, gene-specific readthrough rates, readthrough lengths, inter-gene splicing /chimera read rates at different timepoints, the relationship between readthrough and inter-gene splicing, etc.

Major comments:

- While there is high correlation between biological replicates, (fig. 1b), the numbers deviate in several folds between replicates. There is no explanation as to what may be causing this, what is the reason for this, and how this difference is dealt with.
- Comparison between nanopore and illumina profiles is interesting. Qualitatively, it seems as if the background of the illumina sequencing is much higher than that of the nanopore, when considering intergenic regions. However, the y-axis scale of the peaks is somewhat higher for the illumina. Since the manuscript highlights data processing methodologies as one of its novelties, a more systematic comparison between the illumina and nanopore results is required, including background / intergenic region analysis, and comparison to gene region analysis.
- The authors present an "innovative approach to error-correction that rescues otherwise unmapped reads, improves splice junction mapping, and allows for accurate prediction of encoded proteins.". However, their method relies on illumina short read sequencing of the same RNA, and their ORF prediction basically relies on correction from the reference genome. Therefore, if parallel illumina RNAseq is required, it is unclear what the nanopore sequencing in general, and their error correction method in particular, are good for. If increased coverage is the goal, then simply allowing more mapping errors in the qualification of each read would do. It seems that novel ORFs are not going to come out of the nanopore seq anyway, as the reference genome is what is finally being used. This part needs to be clarified, and illustrated through specific analyses that support the claims. Otherwise, in its current state, it seems obsolete.
- It seems that their method would be more useful when alternative splicing is considered (which is prevalent in the host genome in this case), or to determine splicing efficiency and rates, which are relevant to the viral genome as well. Neither of these are discussed. The authors should relate to both, or at least to the latter – what are the splicing efficiencies as predicted by their method? How does it

compare to the illumina RNAseq-based quantifications? This is critical in order to show the advantages of their method.

- Readthrough is an area of growing interest. While Rutkowski et al has shown that there is no readthrough in the viral transcriptome, the authors claim otherwise. This discrepancy is discussed in the discussion, however, in order to establish their claims, the authors need to show more rigorous analyses of bulk and gene-specific readthrough, as illustrated by their nanopore-seq as well as illumina RNAseq for the chimera transcripts. Additionally, they should download the Rutkowski et al. data and perform a direct comparison of the readthrough patterns observed.
 - o Specifically , what does the Rutkowski et al. data look like with respect to the predicted chimeras? What is the relationship between the authors and Rutkowski et al's data? Would a direct comparison help reconcile the potential discrepancy in their claims?
- Fig. 4: These predictions need to be supported by data: how many reads support each chimera/trans-splicing transcript? What is the % of chimera-reads vs. canonical reads?
- Readthrough and trans-splicing: what are the rates of each in the data? How does readthrough rate correlate to inter-gene splicing ? how frequent is readthrough? How frequent is chimera RNA generation?
- Fig. 5: read density tracks would be very useful here, both for nanopore-seq and illumina RNA-seq.
- Fig. 6a: +CHX sample is only shown for 6h, when the chimera mRNA is not present anyway. Need to include 18h+CHX as well to establish that novel protein synthesis is required for generation of the chimera.
 - o On the same note, is the chimera transcript absent at 6h because of no readthrough? Or lack of inter-gene splicing?
- Regarding the RL2-UL1 chimera in other cell types– I believe that “data not shown” is no longer acceptable in most journals . So the data should either be shown or the statement removed.

Minor comments:

- Typos in the abstract
- Fig. 6b: it is unclear how normalization to 18s rRNA was performed. Usually, normalizer gene Ct values can be subtracted from the actual gene. If so, this means that the actual Ct values of the red chimera are actually much higher than 30. Often, above 33-34 the Ct values are considered unreliable. In that case, what are the raw values? At least the late timepoint Ct values need to be lower than the 30 range. Also, in such a range the negative controls need to be presented.

Reviewer #2:

Remarks to the Author:

Depledge et al. present a detailed study of HSV-1 RNAs using native RNA sequencing on nanopore arrays and they demonstrate that the HSV-1 transcriptome is more complex than previous RNA-seq studies have shown. The authors compare their method, which directly sequences single RNA molecules without RT-PCR as required for other RNA-seq methods and state that their method is without bias. The manuscript is very technical in the description of the methodology, validation, corrections and bioinformatics analysis. The studies reveal new transcription start sites, read through terminations and new cis-spliced RNAs not previously seen with other methods. The authors also demonstrate a fused transcript between ICP0, an immediate protein encoding a ubiquitin ligase and glycoprotein L, a late protein that is important for HSV-1 fusion during entry. It is also shown that the predicted protein product is expressed as determined by western blot analysis. Several other fused transcripts that could lead to other fusion proteins are also shown at the RNA level but not at the protein level. The manuscript is written in a very technical style making it somewhat difficult to appreciate the advances described in the work. There are several concerns that the authors should address:

1. Although it is clear that nRNA-seq has revealed additional complexity of the HSV-1 transcriptome, it is less clear what the significance of these findings will be to the field.
2. While the previously unrecognized transcripts have been uncovered by this approach, it is not clear if or what these transcripts contribute to infection.
3. Although the fusion of ICP0 and gL was demonstrated to produce protein, it is not clear what significance this fusion protein might have to infection. It is a late gene product, although ICP0 is an early gene product and at late times, ICP0 is found in the cytoplasm, whereas, gL is processed in the ER and Golgi and comprises part of the virion envelope. It has previously been shown that gC, another HSV-1 glycoprotein expresses an alternatively spliced RNA that is translated to a secreted protein that comprises ~5% of the total gC expressed. However, no role for secreted gC has been revealed.
4. The authors should focus the discussion more on the potential significance of the data that they present as the introduction and results emphasize the technical merits of this approach.

Reviewer #3:

Remarks to the Author:

In the manuscript "Native RNA sequencing on nanopore arrays redefines the transcriptional complexity of a viral pathogen" by Depledge et al., the authors describe their work in using native RNA sequencing on HSV-1. This is really interesting work, and beyond a preprint on flu virus (<https://www.biorxiv.org/content/early/2018/05/08/300384>) the first work I'm aware of directly looking at viral genomes with an intervening RT step. The authors leverage this well, characterizing different aspects of the virus, even identifying novel protein fusions and demonstrating that they actually result in a translated protein.

My biggest concern is in how they were correcting their nanopore data. I'm not sure their approach is ideal, especially for some of the potential applications. That said, I think their `_specific_` method for correction is fine for looking at broad changes like splicing changes or transcription start/end locations.

Specific comments:

1. Typo in abstract: Line 40 "phow" should be "how"
2. More details on how the data in the right panel of Supplemental Figure 1a were classified would be helpful. Especially as it seems that ~5% of the reads from the null virus sample were classified as virus - does this make sense in context? More background on the expected behavior of dVHS would be useful.
3. It seems inaccurate to call the correlation between Illumina and nanopore abundances "similar but not identical profiles", given the *low* R^2 values measured in Supplemental Figure 1c. The differences certainly make sense, especially in light of Figure 1c, but a different examination looking at specific areas instead of across the entire viral genome could be useful, or perhaps a measure of the correlation versus the sliding window. This is pretty important, as it is necessary to allow connection to the body of work using Illumina/short-read methodologies.
4. Supplementary Figure 2 - it isn't clear that there is much value added (at least for the virus) to use more Illumina data for correction. Some sort of quantification in addition to the violin plots could be helpful - the median error rate, stratified for insert/deletion/mismatch per read and how that drops with more data could allow a threshold of the amount of data needed to be set. This kind of practical information will help the field to design experiments appropriately.

5. Fig 2b - it is not clear that the minimum cigar length versus alignment is the "best" alignment. 1) CIGAR does not take into account mismatches, which are less of a problem in circular consensus sequencing on pacbio, certainly, but remain an issue for both raw pacbio and raw nanopore sequencing data 2) What if the reference sequence is wrong? A better way to measure this would be to evaluate the percentage of bases accurately matched per read, along with the percentage of insertion, deletion, mismatch as a fraction of the total read length.
6. It is confusing to me why *more* reads result in a higher CIGAR length in Fig 2b? It seems from the figure that you subsampled more reads and got a longer CIGAR. This gives further credence that CIGAR length alone does not optimally estimate accuracy.
7. I also take issue with the idea of using an alignment to a "reference" to establish and correct the sequence (Fig 2a 3rd part). It seems to me this is reliant on having an accurate reference sequence, but for viruses, with the many viral quasispecies and the rapid rate of mutation, you would just be removing some of this natural diversity to match an existing reference strain.
8. It also seems that sequencing errors at or near the internal splice sites could confound the generation of pseudo-transcripts. Do you appear to generate any false pseudo-transcripts using this method? And if so, please briefly discuss how this may impact transcript identification.
9. Labeling of "Read Depth on Figure 3a is misleading, since it is technically the 5' of the read (pTSS). It is also not addressed in the legend the difference between the top and bottom sets of tracks, though I assume from context one is + strand and the other - strand.
10. It would be nice to see an example of IGV data that shows read-through transcription.
11. The authors gloss over that some genes (example the two exon grey gene at ~125kb on the + strand) do not seem expressed at all? Is this expected based on the time of sampling?
12. The quantities of the genes also appear to alter over time, as do the uses of the alternative start sites. Quantification of this and the correlation of these results with Illumina data might be useful to demonstrate the accuracy of quantitation, especially if molecular indexing is used with Illumina sequencing, and 5' ends should be measurable for quantitation as was done earlier in the paper for entire parts of the RNA.
13. Suggest legend for fig 1a left to indicated the grey scale meanings instead of just in the figure legend
14. What is the frequency of the spliced/fusion ORFs compared to the normal unspliced ORFs, especially versus time? UL1-RL2 is explored in Figure 6, but at least the data for the other splice variants from the nanopore sequencing should be plotted/presented?
15. It would be helpful to provide a repository of scripts used for the analysis, especially given the new method of analysis developed here.

Reviewer #1 (Remarks to the Author):

Point 1. The study presents the generation and analysis of long read RNA-seq using nanopore arrays, in order to examine the transcriptome of the HSV1 virus following infection. The authors also generate parallel illumina short-read mRNA-seq for the same samples and present a novel method for improvement of mapping and splice junction assignment. Finally, they show that readthrough transcription occurs post HSV1 infection, and that subsequent novel splicing events within readthrough RNAs lead to chimera RNAs and proteins, one of which could be detected by WB.

Identification of the UL2-UL1 chimeric mRNA was entirely unexpected because this region of the viral genome has been studied extensively. It is not clear to us what new information a statistical analysis will provide given that we can readily detect the ICP0-gL fusion product in lysates from wild type HSV-1 infected cells. However, we have expanded the analysis to validate the UL52-54 read-through/splice by end-point PCR.

Point 2. The findings of the authors regarding readthrough and subsequent inter-gene splicing leading to chimera proteins are very interesting. However, given the current state of analyses, they remain anecdotal. No rigorous statistics or data is presented for readthrough, inter-gene splicing, or both, not even for the single RL2-UL1 chimera. As figure 6 is very convincing, especially the WB, it is highly likely that these chimeras exist, and result from readthrough. This is a major novelty factor, however without more rigorous analyses one cannot really establish its prevalence in the data, and the biological impact. A thorough comparison to illumina RNA-seq should be made in that regards, as well as to previously published nascent RNA-seq by Rutkowski et al.

Even though both studies used similar viruses to infect primary human fibroblasts, it is difficult to make meaningful comparisons to the Rutkowski study, which analyzed nascent RNAs. As discussed below, the chief issue is that the majority of viral read-through transcripts are detected at 18 hpi, which is later in the infection cycle than was analyzed by Rutkowski et al (1-8 hpi). As the reviewer notes, the fact that we can readily detect the ICP0-gL fusion protein in infected cell lysates opens the possibility that it has biological impact.

Point 3. The manuscript figures are mainly genome tracks. These are good as illustrations but should 1

accompany figures illustrating general statistics of the data to support the authors claims. This is true with regards to splicing rates, general readthrough rates, gene-specific readthrough rates, readthrough lengths, inter-gene splicing /chimera read rates at different timepoints, the relationship between readthrough and inter-gene splicing, etc.

We have added a number of more quantitative analyses but also feel that are significant advantages to using genome tracks to illustrate the data including comparisons of nanopore and Illumina sequencing outputs. As noted in the manuscript and in other answers here, many of the metrics mentioned by the reviewer (e.g. read-through rates and lengths) cannot be accurately determined from this study without purposefully sequencing pre-mRNAs.

Major comments:

Point 4. While there is high correlation between biological replicates, (fig. 1b), the numbers deviate in several folds between replicates. There is no explanation as to what may be causing this, what is the reason for this, and how this difference is dealt with.

Like any sequencing methodology, direct RNA-Seq runs yields different numbers of sequencing reads, even if runtime (18 h) and total RNA input (25 micrograms) are held constant. The number of reads may be influenced by both technical and biological elements such as the efficiency of poly(A) selection (technical) and the selective action of HSV-1 vhs in degrading host and viral mRNAs (biological). What is important is that even when biological replicates yield differing numbers of reads (here the difference is slightly less than 2-fold), the relative abundances of viral (Fig 1b – left) and host (Fig 1b – right) transcripts do not markedly differ. We have modified the figure legend to describe this more effectively.

Point 5. Comparison between nanopore and illumina profiles is interesting. Qualitatively, it seems as if the background of the illumina sequencing is much higher than that of the nanopore, when considering intergenic regions. However, the y-axis scale of the peaks is somewhat higher for the illumina. Since the manuscript highlights data processing methodologies as one of its novelties, a more systematic comparison between the illumina and nanopore results is required, including background / intergenic region analysis, and comparison to gene region analysis.

To address this, we have created a specific subsection in the results to better examine the differences between Illumina and nanopore sequencing of the same samples (including better normalization between nanopore and Illumina datasets). While some of the suggested analyses are prohibited because of the arrangement of complex transcriptional units across the HSV-1 genome, combined with poor and incomplete gene annotations, we have provided a far greater depth of analysis than any previous study.

Point 6. The authors present an “innovative approach to error-correction that rescues otherwise unmapped reads, improves splice junction mapping, and allows for accurate prediction of encoded proteins.”. However, their method relies on illumina short read sequencing of the same RNA, and their ORF prediction basically relies on correction from the reference genome. Therefore, if parallel illumina RNAseq is required, it is unclear what the nanopore sequencing in general, and their error correction method in particular, are good for. If increased

coverage is the goal, then simply allowing more mapping errors in the qualification of each read would do. It seems that novel ORFs are not going to come out of the nanopore seq anyway, as the reference genome is what is finally being used. This part needs to be clarified and illustrated through specific analyses that support the claims. Otherwise, in its current state, it seems obsolete.

The primary issue with using Illumina sequencing alone in this context is that the compact gene-dense organization of viral genomes (exemplified by HSV-1) results in many distinct transcripts that map to the same gene cluster and these cannot be accurately deconvoluted using short-read Illumina approaches alone. We would argue that direct RNA-Seq, even without error-correction, is extremely useful for generating general overviews (maps) of the transcriptome and for identifying novel transcripts. Error-correction improves this significantly while still retaining the original transcript information in terms of extreme mapping co-ordinates. It also provides accurate splice-site mapping. Therefore, we would argue that the use of nanopore or nanopore+Illumina correction allows for the consideration of one transcript = one mRNA relationship, which is patently not the case with Illumina sequencing alone. We have addressed this in detail within the revised manuscript section entitled "*The utility of error-correction and pseudo-transcripts*".

Point 7. It seems that their method would be more useful when alternative splicing is considered (which is prevalent in the host genome in this case), or to determine splicing efficiency and rates, which are relevant to the viral genome as well. Neither of these are discussed. The authors should relate to both, or at least to the latter – what are the splicing efficiencies as predicted by their method? How does it compare to the illumina RNAseq-based quantifications? This is critical in order to show the advantages of their method.

The drawback to direct RNA sequencing on the MinION is that relatively few host reads are captured (too few for a meaningful analysis of splicing). Add to this the known actions of viral proteins such as vhs and ICP27 and any analysis of host transcripts will be flawed. In this study we focus on the virus and the advantage to nanopore sequencing is that splice site mapping is far more accurate than with Illumina sequencing, principally because longer reads allow more accurate identification. Unfortunately, due to several observed issues with sequencing artefacts masquerading as splice junctions (detailed in the manuscript) there is no reliable way to analyze splicing efficiencies. We have however made note of the relative scarcity of (valid) spliced HSV-1 reads and discuss this in the context of read-through.

Point 8. Readthrough is an area of growing interest. While Rutkowski et al has shown that there is no readthrough in the viral transcriptome, the authors claim otherwise. This discrepancy is discussed in the discussion, however, in order to establish their claims, the authors need to show more rigorous analyses of bulk and gene-specific readthrough, as illustrated by their nanopore-seq as well as illumina RNAseq for the chimera transcripts. Additionally, they should download the Rutkowski et al. data and perform a direct comparison of the readthrough patterns observed. Specifically, what does the Rutkowski et al. data look like with respect to the predicted chimeras? What is the relationship between the authors and Rutkowski et al.'s data? Would a direct comparison help reconcile the potential discrepancy in their claims?

We appreciate and share the reviewers interest in transcriptional read through. Unfortunately, a direct comparison between the two studies is not appropriate given the very different sequencing approaches used

and other differences in experimental design including the duration of the infections. Our study was not designed to comprehensively analyze read-through in viral genomes but is instead geared toward transcript discovery. That said, we have examined the Rutkowski et al. dataset for evidence of low-level read-through (evident in Figure 9 of the Rutkowski et al. paper) and looked for, but did not find, sequence reads corresponding to the novel splice junctions we have discovered. We attribute this to the relative depth of sequencing used in the Rutkowski et al. study and the fact that they sequenced total rather than polyA-selected RNA. Most importantly, they only profiled the first 8 h of infection. Our qPCR analysis showed that the RL2-UL1 transcript fusion is particularly scarce at this time, becoming more abundant later. Although our data suggests that read through transcription of the HSV-1 genome can occur and is clearly not cell line or viral strain specific, the impact on viral biology is probably limited because of the highly compact nature of the genome. There are additional poly(A) signal sites close to the ends of all ORFs or ORF arrays that will most likely result in the appropriate processing of the transcript 3' ends to produce translatable mRNAs. We discuss this in greater detail in the revised discussion.

Point 9. Fig. 4: These predictions need to be supported by data: how many reads support each chimera/trans-splicing transcript? What is the % of chimera-reads vs. canonical reads?

Due to the comparatively low prevalence of chimeric reads, we have addressed this by a time-course RT-qPCR for both the RL2-UL1 and UL52-UL54 fusion.

Point 10. Readthrough and trans-splicing: what are the rates of each in the data? How does readthrough rate correlate to inter-gene splicing? how frequent is readthrough? How frequent is chimera RNA generation?

It is clear from the small percentage of transcripts with 3 or more AAUAAA poly(A) signal (PAS) motifs that read-through rates are comparatively low and do not increase/decrease during the infection period. The preference of HSV-1 for using canonical AAUAAA PAS sites, located within close proximity to the 3' end of the ORF agrees with the Rutkowski et al. observations of sequence features that confer a degree of 'protection' against read-through transcription.

Point 11. Fig. 5: read density tracks would be very useful here, both for nanopore-seq and illumina RNA-seq.

As part of the reconfigured manuscript, we have included Integrative Genomics Viewer (IGV) screenshots as a supplementary figure to better exemplify the (low level) read-through that allows the generation of chimeric transcripts.

Point 12. Fig. 6a: +CHX sample is only shown for 6h, when the chimera mRNA is not present anyway. Need to include 18h+CHX as well to establish that novel protein synthesis is required for generation of the chimera.

We agree and have now performed this later time point (see new Fig. 6c and 7b).

Point 13. On the same note, is the chimera transcript absent at 6h because of no read-through? Or lack of inter-gene splicing?

This is an interesting question as it might more shed light on the underlying mechanisms of read through. The splice site within the chimeric transcript is detectable at 6 hpi in the RT-qPCR data indicating that both read-through and subsequent splicing are occurring. However, we do not observe the chimeric transcript in nanopore data at 6 hpi but this is not too surprising given the (comparatively) limited depth of sequencing that can be achieved.

Point 14. Regarding the RL2-UL1 chimera in other cell types– I believe that “data not shown” is no longer acceptable in most journals. So, the data should either be shown or the statement removed.

The reviewer is correct. We have included PCR data (new Fig. 7b) confirming the usage of the splice junction that gives rise to the RL2-UL1 chimera in both neurons and ARPE-19 retinal epithelial cells. Furthermore, we have also expanded our analysis to the splice site that gives rise to the chimeric transcript UL52-54 (new Fig. 6c).

Minor comments:

- Typos in the abstract

This has been corrected.

- Fig. 6b: it is unclear how normalization to 18s rRNA was performed. Usually, normalizer gene Ct values can be subtracted from the actual gene. If so, this means that the actual Ct values of the red chimera are actually much higher than 30. Often, above 33-34 the Ct values are considered unreliable. In that case, what are the raw values? At least the late timepoint Ct values need to be lower than the 30 range. Also, in such a range the negative controls need to be presented.

We have modified the text of both the Experimental Methods and figure legend to explain this better and have also included the Ct values without normalization in new Fig. 6b and 7c. We also present values for the spliced canonical RL2 and unspliced UL1, UL52 and UL54 transcripts. While we agree the high Ct values indicate low abundance transcripts, there is clearly sufficient RL2-UL1 mRNA for us to readily detect the protein product by immunoblotting.

Reviewer #2 (Remarks to the Author):

Depledge et al. present a detailed study of HSV-1 RNAs using native RNA sequencing on nanopore arrays and they demonstrate that the HSV-1 transcriptome is more complex than previous RNA-seq studies have shown. The authors compare their method, which directly sequences single RNA molecules without RT-PCR as required for other RNA-seq methods and state that their method is without bias. The manuscript is very technical in the description of the methodology, validation, corrections and bioinformatics analysis. The studies reveal new transcription start sites, read through terminations and new cis-spliced RNAs not previously seen with other methods. The authors also demonstrate a fused transcript between ICP0, an immediate protein encoding a ubiquitin ligase and glycoprotein L, a late protein that is important for HSV-1 fusion during entry. It is also shown that the predicted protein product is expressed as determined by western blot analysis. Several other fused transcripts that could lead to other fusion proteins are also shown at the RNA level but not at the

protein level. The manuscript is written in a very technical style making it somewhat difficult to appreciate the advances described in the work. There are several concerns that the authors should address:

We apologize for the technical style, but as we are sure the reviewer appreciates, this is difficult to avoid when presenting a new and complicated technology. With this in mind we have reorganized the manuscript such that the less biological (more informatic) aspects are concentrated in the supplemental figures/files and in the methods. We have also re-written many sections make them more accessible to all readers with the expectation that other investigators will wish to apply nanopore sequencing and the associated analyses for their own transcriptomic studies.

Point 1. Although it is clear that nRNA-seq has revealed addition complexity of the HSV-1 transcriptome, it is less clear what the significance of these findings will be to the field.

A challenge for the field is that we lack a comprehensive map of the viral transcriptome. A lot of the information on the HSV-1 transcriptome is anecdotal (i.e. not gathered in a systematic fashion) or is based on the identification of ORFs without a clear understanding of how these relate to individual mRNAs. This is true for other herpesviruses (HCMV, PRV, VZV, MHV68 and KSHV) and have seen similar systematic attempts to map out the transcriptomes at high resolution. The new details force us to re-examine older data including expression analyses that may not adequately distinguish between different transcripts or the impact of genetic manipulations. This is most relevant where multiple distinct transcripts are shown to span a gene cluster or even an intergenic region. The work also raises the important question of whether read-through transcription is simply a by-product of how the virus targets the host transcriptome or whether it has evolved as a more deliberate mechanism to expand the coding potential of the viral genome.

Point 2. While the previously unrecognized transcripts have been uncovered by this approach, it is not clear if or what these transcripts contribute to infection.

The same could be said of other HSV-1 transcripts which are poorly characterized. Clearly there is a requirement to determine the function / importance of these transcripts but this is beyond the scope of this first study.

Point 3. Although the fusion of ICP0 and gL was demonstrated to produce protein, it is not clear what significance this fusion protein might have to infection. It is a late gene product, although ICP0 is and IE and at late times, ICP0 is found in the cytoplasm, whereas, gL is processed in the ER and Golgi and comprises part of the virion envelope. It has previously been shown that gC, another HSV-1 glycoprotein expresses an alternatively spliced RNA that is translated to a secreted protein that comprises ~5% of the total gC expressed. However, no role for secreted gC has been revealed.

Naturally, we are aggressively pursuing a variety of experiments to identify the function of the ICP0-gL fusion as well as its localization and association with host and viral proteins. As noted, the conservation of the splice sites across many HSV-1 strains suggests some selective pressure to maintain expression of either the chimeric mRNA or protein. In the short term, identification of this fusion prompts reevaluation of some previous observations relating to both the localization and functions of ICP0 and gL towards the end of the infection cycle.

Point 4. The authors should focus the discussion more on the potential significance of the data that they present as the introduction and results emphasize the technical merits of this approach.

We have tried to do this as much as possible throughout the revised manuscript. Hopefully the significance of the new information concerning polyadenylation signal usage and its relationship to read-through is more apparent.

Reviewer #3 (Remarks to the Author):

In the manuscript “Native RNA sequencing on nanopore arrays redefines the transcriptional complexity of a viral pathogen” by Depledge et al., the authors describe their work in using native RNA sequencing on HSV-1. This is really interesting work, and beyond a preprint on flu virus (<https://www.biorxiv.org/content/early/2018/05/08/300384>) the first work I’m aware of directly looking at viral genomes with an intervening RT step. The authors leverage this well, characterizing different aspects of the virus, even identifying novel protein fusions and demonstrating that they actually result in a translated protein.

We are delighted the reviewer recognizes the cutting-edge nature of this study. This underscores the need to perform critical evaluations of reproducibility and maintenance of RNA quality. As we have argued in a recent review (Depledge et al. 2018 J. Virology, in press), we believe native RNA sequencing and this technology in particular, will be of great value in studies of intracellular pathogens with gene-dense genomes.

My biggest concern is in how they were correcting their nanopore data. I’m not sure their approach is ideal, especially for some of the potential applications. That said, I think their `_specific_` method for correction is fine for looking at broad changes like splicing changes or transcription start/end locations.

We have confidence in our error-correction approach but acknowledge the reviewer’s concern that it may not be applicable to all applications of nanopore sequencing. We have addressed this in the discussion, making the point that it is not an optimal approach for dealing with small RNA viruses. Nonetheless, for aims as broad as interrogating transcript initiation, splicing patterns, read through transcription, and for transcript discovery, its utility remains high for studies of DNA viruses as well as unicellular organism such as bacteria and parasites.

Specific comments:

Point 1. Typo in abstract: Line 40 “phow” should be “how” This is corrected.

Point 2. More details on how the data in the right panel of Supplemental Figure 1a were classified would be helpful. Especially as it seems that ~5% of the reads from the null virus sample were classified as virus - does this make sense in context? More background on the expected behavior of dVHS would be useful.

The legends to Figure 1 and Supplemental Figure 1 are amended to better explain the classification. Note however that we do not have a null-virus sample, rather this sample was generated by infecting with a virus in which a single ORF (vhs) is deleted. As vhs is critical to enabling host shutoff (degradation of mRNAs),

infections proceed much slower and the proportion of viral mRNAs is therefore much less. New text at the end of the first results section makes this clear.

Point 3. It seems inaccurate to call the correlation between Illumina and nanopore abundances “similar but not identical profiles”, given the *low* R^2 values measured in Supplemental Figure 1c. The differences certainly make sense, especially in light of Figure 1c, but a different examination looking at specific areas instead of across the entire viral genome could be useful, or perhaps a measure of the correlation versus the sliding window. This is pretty important, as it is necessary to allow connection to the body of work using Illumina/short-read methodologies.

To address this, we have created a specific subsection in the results to better examine differences between Illumina and nanopore sequencing of the same samples (including better normalization between nanopore and Illumina datasets). While some of the suggested analyses are prohibited by the arrangement of mono- and poly-cistrons along the HSV-1 genome, combined with poor and incomplete gene annotations, we have provided a far greater depth of analysis than previously.

Point 4. Supplementary Figure 2 - it isn't clear that there is much value added (at least for the virus) to use more Illumina data for correction. Some sort of quantification in addition to the violin plots could be helpful - the median error rate, stratified for insert/deletion/mismatch per read and how that drops with more data could allow a threshold of the amount of data needed to be set. This kind of practical information will help the field to design experiments appropriately.

We observed several interesting features through our subsampling approach, one of which was that while there was a general trend toward shortening of CIGAR string lengths as the number of Illumina reads used was increased, this was an overall effect and not one necessarily observed when inspecting a given read (i.e. more data does not necessarily mean better correction). This is exemplified in Figure 2b (read_003) where the ‘optimal’ correction (as defined by the shortest CIGAR string) was not observed when correcting with the largest number of Illumina reads. This is what informed our decision to use a decision matrix, filtering for the ‘best’ possible correction through a range of possibilities on a per read basis. We have expanded our description of the error correction approach in the methods.

Point 5. Fig 2b - it is not clear that the minimum cigar length versus alignment is the “best” alignment. 1) CIGAR does not take into account mismatches, which are less of a problem in circular consensus sequencing on PacBio, certainly, but remain an issue for both raw PacBio and raw nanopore sequencing data 2) What if the reference sequence is wrong? A better way to measure this would be to evaluate the percentage of bases accurately matched per read, along with the percentage of insertion, deletion, mismatch as a fraction of the total read length.

The reviewer is correct that mismatches can be problematic but we can overcome this by generating what we term ‘pseudo-transcripts’, in which we use the mapping co-ordinates of the error-corrected read to identify the 5' and 3' ends of transcripts along with internal information on splice junctions. This step is also critical

because error-correction, while improving the mapping accuracy of reads, does not remove all instances of mismatches and/or internal indels – however by significantly reducing them, we are much more confident in our ability to robustly generate pseudo-transcripts that reflect *in vivo* polyadenylated RNAs. While we don't condone applying this error-correction approach to RNA viruses, dsDNA viruses and higher order pathogens such as bacteria have far more stable genomes and in many cases have been sequenced sufficiently to validate existing reference genomes. Given that viral genomes are compact, and most genes have only single exons, we are confident that the minimal cigar string lengths are representative of the best possible alignments. We have however split Supplementary Figure 2 into two separate figures (one for the virus, one for the host), and included additional panels showing how well error-correction performs in terms of reducing the number of transcripts with insertions or deletions.

Point 6. It is confusing to me why *more* reads result in a higher CIGAR length in Fig 2b? It seems from the figure that you subsampled more reads and got a longer CIGAR. This gives further credence that CIGAR length alone does not optimally estimate accuracy.

We agree with the reviewer that this is a surprising observation but we don't see why this diminishes the choice of CIGAR string length as a measure of correction effectiveness. We suspect this is a feature of how proofread performs its consensus calculations when performing the 'correction' step following Illumina mapping to nanopore reads. Importantly, it emphasizes the importance of our implementation of a decision matrix to identify the optimally-corrected version of each read (from the point of view of subsequently generating the pseudo-transcript).

Point 7. I also take issue with the idea of using an alignment to a "reference" to establish and correct the sequence (Fig 2a 3rd part). It seems to me this is reliant on having an accurate reference sequence, but for viruses, with the many viral quasi-species and the rapid rate of mutation, you would just be removing some of this natural diversity to match an existing reference strain.

We understand this concern and have included several additional statements emphasizing how this error-correction strategy is only useful for agents (viral or otherwise) with stable and (relatively) slowly evolving genomes. For HSV-1, like most other large dsDNA viruses, the mutation rate is known to be very low and as such they do not exist as quasi-species. We do not advocate this error-correction methodology for all organisms and we have included statements recognizing its utilities and limitations.

Point 8. It also seems that sequencing errors at or near the internal splice sites could confound the generation of pseudo-transcripts. Do you appear to generate any false pseudo-transcripts using this method? And if so, please briefly discuss how this may impact transcript identification.

This is a good question. As is detailed in Figure 3D, we determined that our ability to predict ORFs within transcripts was substantially improved by error-correction. While it is certainly possible that sub-optimal error-correction may impact on the accurate identification of splice junction junctions, the improvement observed would argue that any erroneous transcripts are in the minority. While a full splicing analysis would help to

address this, we were unable to do so because of 'splicing artefacts' in a proportion of nanopore reads (detailed in the methods section of the manuscript).

Point 9. Labeling of "Read Depth on Figure 3a is misleading, since it is technically the 5' of the read (pTSS). It is also not addressed in the legend the difference between the top and bottom sets of tracks, though I assume from context one is + strand and the other - strand.

We have changed the labelling to 'Read Count' and also addressed + and – strand comment in the legend the new Fig 4.

Point 10. It would be nice to see an example of IGV data that shows read-through transcription.

Excellent suggestion. This is now included as a new Supplementary Fig. 8.

Point 11. The authors gloss over that some genes (example the two exon grey gene at ~125kb on the + strand) do not seem expressed at all? Is this expected based on the time of sampling?

We have amended the text in the section entitled 'Profiling viral transcription at early and late stages of infection' to clarify that mRNAs mapping to all canonical viral ORFs are detected but that expression levels vary widely which relates to factors such as gene expression kinetics and mRNA stability.

Point 12. Suggest legend for fig 1a left to indicated the grey scale meanings instead of just in the figure legend

Figure 1a already has a legend shown within the body of the figure. We think the reviewer means old Fig. 4a and have added a legend to show gray scale meanings here.

Point 13. What is the frequency of the spliced/fusion ORFs compared to the normal unspliced ORFs, especially versus time? UL1-RL2 is explored in Figure 6, but at least the data for the other splice variants from the nanopore sequencing should be plotted/presented?

We have refocused the manuscript on the UL1-RL2 and UL52-54 fusion transcripts as these are the most robustly detectable. The frequency of fusion transcripts is comparatively low, as is now illustrated by the specific qPCR components added to the final figures. We favor this approach due to the relatively low numbers of reads currently obtained by nanopore sequencing on the MinION platform.

Point 14. It would be helpful to provide a repository of scripts used for the analysis, especially given the new method of analysis developed here.

This is provided in Supplementary File 2.

Reviewers' Comments:

Reviewer #1:

Remarks to the Author:

I have read the revised manuscript, and many of my comments have not been addressed. I understand now, which was not clear in the initial version of the manuscript, that inter-transcript splicing is an extremely rare event, so rare that it is supported by only 1 read for the RL2-UL1 chimera, and 3 reads for the UL52-UL54 chimera, and this is why no statistics can be provided on such low numbers.

Readthrough is not evident at all from the data for the RL2-UL1 transcript (last sup figure), while it is evident for the UL52-UL54 transcript, but its extent is not quantified anywhere. Additionally, the readthrough in the latter region seems to be very short (panel b), while for the former, it should have been >2.5kb long. It therefore seems unlikely to me that the RL2-UL1 chimera would result from readthrough, as there is no single read to support this. Furthermore, its abundance in qPCR is extremely low at all timepoints, and no negative control is provided albeit my request. It is therefore hard to determine whether this qPCR product is real.

Thus, there is a gap between the IP experiment in Fig. 7C and the amount of evidence that lead to the conclusion that this transcript exists. To fill in this gap the authors need to provide multiple replicated for Fig. 7C, and provide additional evidence for the existence of this protein and its function.

Additionally, it is clear to me that the authors did not want to compare their data to the Rutkowski et al. , not even at the 6h timepoint to show reproducibility. Also, they did indicate that they observe readthrough earlier on in a specific location, and thus this at least should have been compared to Rutkowski et al. . Also, if there is readthrough, short reads from illumina should confirm this, and no such analysis has been provided.

I think that the data has some major drawbacks, and thus the authors couldn't address many of the comments, and didn't address some others that they could (for example simply quantifying the extend of readthrough out of total transcript regions in the UL52-UL54 case). I think that in that case they should have backed their findings by some more experiments, to compensate for the lack of responses to the comments made, but I didn't find anything new.

Finally, I understand there is a technical issue in one of their replicates, which deeply confounds the identification of splicing products. Since their identification of novel spliced chimeras result from 1-3 spliced reads, I think this technical issue is a major problem. They should have at least repeated these runs again to see that this does not confound the results.

Reviewer #2:

Remarks to the Author:

This is a revised manuscript by Depledge et al. who present a detailed study of HSV-1 RNAs using native RNA sequencing on nanopore arrays. The authors demonstrate that the HSV-1 transcriptome is more complex than previous RNA-seq studies have shown in part because the HSV-1 genome is dense and compact and the short reads from Illumina sequencing have missed some of this complexity. The authors compare their method, which directly sequences single RNA molecules without RT-PCR as required for other RNA-seq methods and state that their method is without bias. In my previous review I was concerned that the manuscript was very technical in the description of the methodology, validation, corrections and bioinformatics analysis making it hard to appreciate the significance of the novel and potentially exciting findings using this new approach. The authors have responded very well

to these concerns and have placed much of the technical description on the supplementary material. The text of the manuscript has been significantly revised and it is now clear what has been done, what has been found and what this means for HSV-1 expression during infection. The studies reveal new transcription start sites, read through terminations and new cis-spliced RNAs not previously seen with other methods. The authors also demonstrate a fused transcript between ICP0, an immediate protein encoding a ubiquitin ligase and glycoprotein L, a late protein that is important for HSV-1 fusion during entry. It is also shown that the predicted protein product is expressed as determined by western blot analysis. While the biological function of this transcript and the translated fused protein has not yet been revealed, the authors state in the response to the reviewers that they are working on determining its role in late infection.

In summary, the authors have significantly modified the manuscript and have addressed all of my concerns in full. To my mind they have also addressed the concerns of the other reviewers satisfactorily also. I have no further concerns.

Reviewer #3:

Remarks to the Author:

I'm largely satisfied with the response, with the exception of the two points below, from original points 3 and 4 through 6 (merged)

3. So, are the authors saying that there is poor correlation between nanopore and Illumina sequencing because Illumina sequencing in this area is inaccurate/non-quantitative due to overlapping transcripts sharing the same RNA cleavage sites? It probably makes sense to bring Fig 2a back in, and perhaps to show correlation between mean read depth in the 100 bp sliding window across genic regions?

4-6. I'm sorry, I still don't understand why additional Illumina data doesn't either correct further or saturate. The authors have not satisfactorily either shown **why** extra Illumina data causes fewer errors to be corrected, or shown using a different error correction method (i.e. pilon or some other tool) that the same behavior is observed. And again, I think this is depending on an accurate reference sequence, meaning that the technique has the potential to miss new changes because of the push to optimization of correction. The authors, in my opinion, are mistaking the removal of indels vs reference with a "more correct" result. This is presupposing the reference is representative of the sample. If a "real" gap exists in the sample versus the reference, the authors would strive to eliminate this via optimization of Illumina data chosen.

Reviewer #1 (Remarks to the Author):

I have read the revised manuscript, and many of my comments have not been addressed. I understand now, which was not clear in the initial version of the manuscript, that inter-transcript splicing is an extremely rare event, so rare that it is supported by only 1 read for the RL2-UL1 chimera, and 3 reads for the UL52-UL54 chimera, and this is why no statistics can be provided on such low numbers. Readthrough is not evident at all from the data for the RL2-UL1 transcript (last sup figure), while it is evident for the UL52-UL54 transcript, but its extent is not quantified anywhere. Additionally, the readthrough in the latter region seems to be very short (panel b), while for the former, it should have been >2.5kb long. It therefore seems unlikely to me that the RL2-UL1 chimera would result from readthrough, as there is no single read to support this.

It seems that there is a misunderstanding regarding the data and some additional clarification is needed. The screenshots in Supplementary Figure 8 are not comprehensive simply because there are too many reads mapping to these loci to show in a single panel. We acknowledge that read-through appears rare in HSV-1 but not to the extent the reviewer is inferring. With regard to the reviewer's request for a determination of read-through frequency - our approach does not allow for a comprehensive analysis of this as we only consider the poly(A) fraction of total RNA (unlike in the Rutkowski et al paper). While we are interested in following up on this in future studies, this would require a different approach that is beyond the aim of this study. Here we are simply evidencing that read-through transcription can occur in the viral genome. In addition, we now specifically point out significant technical differences between the experiments performed here and those in the Rutkowski et al 2015 paper (this is stated on lines 390-430). Notably, our use of poly(A)-selection means that only read-through transcripts that are polyadenylated are captured and also we will tend to miss intronic regions (e.g. RL2-UL1) as we generally capture mature mRNAs. The comment that 'readthrough in the latter region seems to be very short' appears to reflect a concern about viral genomes differing from cellular genomes. As we explicitly state in the discussion (lines 401 - 403), the compact nature of the viral genomes means that read-through transcription will look very different because the higher density of poly(A) signal sites means that most transcripts are likely still polyadenylated. This is in stark contrast to the human genome where it is posited that the RNA pol II transcription is not terminated correctly (no polyadenylation). The reviewer seems to reject the possibility that the RL2-UL1 chimera results from read-through (but doesn't quarrel with UL52-54) but fails to provide an alternative possibility that would explain the fact that we can readily detect its distinctive protein product.

Furthermore, its abundance in qPCR is extremely low at all timepoints, and no negative control is provided albeit my request. It is therefore hard to determine whether this qPCR product is real.

With all due respect to the reviewer, we disagree with this criticism. RNA from uninfected NHDFs was used as a negative control and probing for the viral transcripts yielded the Ct values of greater than 38, consistent with no-detection of specific products. The primers were carefully designed to minimize homology to the human transcriptome and this is confirmed by the lack of specific product in the control samples (Figure 6 b&c). Thus, we believe we have satisfied the reviewer's request for a negative control by accepted criteria.

Thus, there is a gap between the IP experiment in Fig. 7C and the amount of evidence that lead to the conclusion that this transcript exists. To fill in this gap the authors need to provide multiple replicated for Fig. 7C, and provide additional evidence for the existence of this protein and its function.

We provide multiple lines of evidence that the RL2-UL1 chimeric transcript produces a polypeptide with the expected properties in infected cells summarized as follows:

- 1) the apparent Mw of the polypeptide by SDS-PAGE agrees with the size predicted from the chimeric transcript
- 2) The fusion protein contains an epitope that cross-reacts with an ICP0 monoclonal antibody specific for residues 20-105 within the ICP0 N terminus.
- 3) The fusion protein contains an epitope that cross-reacts with an anti-gL monoclonal antibody.
- 4) The fusion protein accumulates within the specific temporal window where the chimeric transcript is produced (at late, but not early times post-infection)

Taken together, these data establish that a single polypeptide of apparent MW of 32 KD containing ICP0 N-terminal and gL C-terminal antigenic determinants exists in HSV-1-infected cells. Between the RT-PCR, qPCR, and Western blotting of raw protein and IP extracts, it is unclear what is missing. We also note that the other reviewers were satisfied with these experiments.

Additionally, it is clear to me that the authors did not want to compare their data to the Rutkowski et al., not even at the 6h timepoint to show reproducibility. Also, they did indicate that they observe readthrough earlier on in a specific location, and thus this at least should have been compared to Rutkowski et al.. Also, if there is readthrough, short reads from illumina should confirm this, and no such analysis has been provided.

This is not correct. In our previous response we confirmed our analysis of the Rutkowski et al 2015 dataset. That said, we now include additional analysis in response to the reviewer's suggestion. Supplementary Figure S8 has been divided into two figures (S8 and S9), incorporating new panels to present coverage plots of the Rutkowski data that clearly show the presence of HSV-1 sequence reads mapping downstream of the RL2 (ICP0) poly(A) signal site. We believe this is definitive evidence of low level readthrough at both 6 and 8 hpi. This observation has been incorporated into the discussion (lines 389 - 391). We hope this will satisfy the reviewer that the Rutkowski et al 2015 study overstated their interpretation that HSV-1 induced read-through does not act on the viral genome based upon their failure to detect readthrough in their dataset.

I think that the data has some major drawbacks, and thus the authors couldn't address many of the comments, and didn't address some others that they could (for example simply quantifying the extend of readthrough out of total transcript regions in the UL52-UL54 case). I think that in that case they should have backed their findings by some more experiments, to compensate for the lack of responses to the comments made, but I didn't find anything new.

Unfortunately, this is a sweeping statement that fails to identify which experiments are required and why they are needed to support or enrich our conclusions. As should be evident from our responses above, we have indeed addressed the reviewer's comments raised in the prior round

and have now done our best to comply with this next round of suggestions. It is noteworthy that the other two reviewers do not share the sentiments articulated by reviewer #1 with respect to our study.

Finally, I understand there is a technical issue in one of their replicates, which deeply confounds the identification of splicing products. Since their identification of novel spliced chimeras result from 1-3 spliced reads, I think this technical issue is a major problem. They should have at least repeated these runs again to see that this does not confound the results.

Our discovery of a previously unknown technical issue with this new technology should not detract from our biological findings, which are supported by repeated experiments and multiple independent methodologies. It is also important not to overblow the minute point of disagreement between the present study and the 2015 Rutkowski et al paper.

The Rutkowski paper concludes that readthrough does not occur on the viral genome based upon their failure to detect it. Importantly, "failure to find" must always be qualified as the event might fall below the limit of detection using a given methodology. Overstating such conclusions by making absolute statements always runs the risk that modified methods may one day reveal precisely what was missed initially. Rather than propagate the absolute statement that viral transcription is immune from read-through, we show using an exciting new technology that this can occur at low level and is potentially important because it produces novel chimeric proteins such as the ICP0-gL fusion.

The concern that spliced reads are only detected at low level becomes irrelevant once a protein product is detected by multiple criteria in HSV-1 infected primary cells. Conclusively showing that a protein product accumulates is the gold standard for evidence that the transcript is in fact decoded into a polypeptide in the correct physiological context.

Reviewer #2 (Remarks to the Author):

This is a revised manuscript by Depledge et al. who present a detailed study of HSV-1 RNAs using native RNA sequencing on nanopore arrays. The authors demonstrate that the HSV-1 transcriptome is more complex than previous RNA-seq studies have shown in part because the HSV-1 genome is dense and compact and the short reads from Illumina sequencing have missed some of this complexity. The authors compare their method, which directly sequences single RNA molecules without RT-PCR as required for other RNA-seq methods and state that their method is without bias. In my previous review I was concerned that the manuscript was very technical in the description of the methodology, validation, corrections and bioinformatics analysis making it hard to appreciate the significance of the novel and potentially exciting findings using this new approach. The authors have responded very well to these concerns and have placed much of the technical description on the supplementary material. The text of the manuscript has been significantly revised and it is now clear what has been done, what has been found and what this means for HSV-1 expression during infection. The studies reveal new transcription start sites, read through terminations and new cis-spliced RNAs not previously seen with other methods. The authors also demonstrate a fused transcript between ICP0, an immediate protein encoding a ubiquitin ligase and glycoprotein L, a late protein that is important for HSV-1 fusion during entry. It is also shown that the predicted protein product is expressed as determined by western blot analysis. While the biological function of this transcript and the

translated fused protein has not yet been revealed, the authors state in the response to the reviewers that they are working on determining its role in late infection. In summary, the authors have significantly modified the manuscript and have addressed all of my concerns in full. To my mind they have also addressed the concerns of the other reviewers satisfactorily also. I have no further concerns.

We thank the reviewer for their very enthusiastic response to our many revisions. In addition, we are also grateful to this reviewer for taking the additional time to comment on our responses to the other reviewer's comments and pointing out that we have satisfied their concerns as well.

Reviewer #3 (Remarks to the Author):

I'm largely satisfied with the response, with the exception of the two points below, from original points 3 and 4 through 6 (merged)

We are pleased the reviewer is satisfied with our many revisions and we apologize for having not provided sufficiently clear responses to his/her original points 3 – 6. We have strived to provide definitive answers to these below.

3. So, are the authors saying that there is poor correlation between nanopore and Illumina sequencing because Illumina sequencing in this area is inaccurate/non-quantitative due to overlapping transcripts sharing the same RNA cleavage sites? It probably makes sense to bring Fig 2a back in, and perhaps to show correlation between mean read depth in the 100 bp sliding window across genic regions?

Precisely, the presence of multiple overlapping transcripts with distinct transcript initiation sites but shared RNA cleavage sites results in different count estimates when using long (nanopore) and short (Illumina) read data. We believe this to be further impacted by the recoding steps inherent in short read RNA sequencing protocols which result in preferential amplification of some transcripts over others. We thank the reviewer for their suggestion and have followed this to generate correlation plots based on 100bp sliding windows across canonically defined genic regions (which account for ~85% of the genome sequence) and have integrated this into Figure 2 panel b.

4-6. I'm sorry, I still don't understand why additional Illumina data doesn't either correct further or saturate. The authors have not satisfactorily either shown *why* extra Illumina data causes fewer errors to be corrected, or shown using a different error correction method (i.e. pilon or some other tool) that the same behavior is observed.

We agree with the reviewer that there is a conceptual difficulty in understanding why increasing the amount of Illumina data used does not continually improve error-correction up to a saturation point. This appears to be related to how proofread operates. Briefly, raw nanopore sequence reads are split into groups containing equal numbers of reads. The reads in each group are subsequently corrected by mapping FLASH-merged Illumina data against all sequence within a group (with each Illumina read being assigned only one best hit). A new consensus sequence is then output for each nanopore read in a given group. We would note that increasing the size of subsampled Illumina datasets does appear to correct further sequence reads mapping to the host as the reviewer would expect. This makes sense given

that human genes are generally singly arrayed with large intergenic regions between them. For the viral genome however the number of overlapping transcripts means that during correction, Illumina sequence reads may be stochastically assigned to the 'wrong' transcript when multiple overlapping transcripts are in the same group. This will obviously impact on final coverage and then influence whether the final consensus sequence is fixed or not.

The optimal theoretical approach (which we are not able to implement) would be to perform error-correction on each nanopore read individually (rather than in groups) but the existing implementations of proofread (or similar mapping-based approaches) are currently too computationally demanding. The reviewer will appreciate that this problem has further scaled with the rapid improvements in nanopore chemistries which are yielding ever greater numbers of reads. While we are hoping in the future to work on a revised version of our error-correction strategy which can better deal with the increase in yield of nanopore reads, that is far beyond the scope of this study.

Alternative error-correction approaches for direct RNA sequencing are sorely lacking at present hence why a custom approach was required. Tools such as Pilon are designed to polish reference-based genome assemblies, which differs markedly from polishing hundreds of thousands of individual nanopore reads where each read represents a distinct RNA. We readily acknowledge that our approach is geared quite specifically to the biological questions we are posing (i.e. this is not a general tool that is appropriate for all direct RNA sequencing projects) and have modified the manuscript accordingly (line 346-350).

Crucially however, we would note that our approach is functional and led us to discover novel transcript structures, which we have confirmed through extensive experimental validation. These structures would have been obscured without error-correction and thus argues for the utility of this approach.

And again, I think this is depending on an accurate reference sequence, meaning that the technique has the potential to miss new changes because of the push to optimization of correction. The authors, in my opinion, are mistaking the removal of indels vs reference with a "more correct" result. This is presupposing the reference is representative of the sample. If a "real" gap exists in the sample versus the reference, the authors would strive to eliminate this via optimization of Illumina data chosen.

The point raised by the reviewer regarding the reference genome is confusing. Our error-correction strategy functions by utilizing Illumina RNA-Seq reads to directly correct nanopore direct RNA-Seq reads in a reference-independent manner. Both sets of reads originate from the same original RNA source. We think the reviewer is asking about the pseudo-transcript generation step and is asking whether indels remaining in the error-corrected transcripts are being incorrectly erased during the pseudo-transcript conversion step. This could in theory impact our efforts to translate pseudotranscripts to identify encoded ORFs. However, as is shown in Figure 3d, our ability to detect intact ORFs increases markedly following error-correction. As we acknowledge however, this approach to error-correction does limit its utility to larger viruses (e.g. herpesviruses) that are not present as swarms/quasispecies. Certainly, this approach would be unsuited to smaller RNA viruses (as we outline in the discussion) which we think is what the reviewer is intimating.

Reviewers' Comments:

Reviewer #3:

Remarks to the Author:

Reviewer 3:

3. Ok, I'm satisfied here.

4-6. I'm still unsatisfied with the language the reviewers are using in the actual manuscript. Though it is clear they understand the limitations of their error correction method, this does not come through in the paper. The line "However, while the generation of psudeo-transcript significant improves our data, it is unlikely to be suitable for applications beyond the analysis of transcript isoforms", does not clearly illustrate the flaws associated with the mis-assignment of Illumina reads at (apparently) greater frequency with greater coverage. Text describing this should be added in the results section (near lines 170-180 I suggest).

4-6b. You are using a "reference" genome - you are generating a CIGAR string by aligning against a reference, then checking for the shortest CIGAR - this is your measurement of "accuracy" - and you are then striving to minimize this by picking the amount of Illumina data which gives the shortest CIGAR. But my point is that you are *assuming* that your reference gives the best answer, and possibly discarding a longer CIGAR that reveals more of the ground truth, but has a higher disagreement with the reference you are using. Essentially you have developed a method on an unknown and have not tested it on a "known" set.

Is it not possible to demonstrate that this error-correction works using a synthetic set that matches or doesn't match a reference to known degrees?

As requested, I also read through Reviewer 1's comments, responses to that:

In response to the first set of comments focusing on read through/inter-transcript splicing - in my opinion the qPCR is validating, but text should be added pointing out the low level of this read-through.

Importantly, the authors provide evidence of the read-through transcript's translation product via an SDS-PAGE analysis, and as long as presented with appropriate caveats in the discussion (noting the lack of replication/small N), it is reasonable to present these results. Their analysis finding this read-through in the Illumina data generated by Rutkowski et al further validates these results.

REVIEWERS' COMMENTS:

Reviewer #3 (Remarks to the Author):

Reviewer 3:

3. Ok, I'm satisfied here.

We thank the reviewer

4-6. I'm still unsatisfied with the language the reviewers are using in the actual manuscript. Though it is clear they understand the limitations of their error correction method, this does not come through in the paper. The line "However, while the generation of pseudo-transcript significant improves our data, it is unlikely to be suitable for applications beyond the analysis of transcript isoforms", does not clearly illustrate the flaws associated with the mis-assignment of Illumina reads at (apparently) greater frequency with greater coverage. Text describing this should be added in the results section (near lines 170-180 I suggest).

We agree with the reviewer and have endeavored to improve the language regarding the limitations of the error-correction method utilized here (see lines 176 (results) and 358 (discussion). Specifically, we are making the point that the generation of pseudo-transcripts, while required to examine the coding capacity of a transcript, eliminates the possibility of examining SNP (e.g. RNA editing) and indel mutations as this information is lost at this step (although that can still be obtained from both raw and error-corrected nanopore reads).

4-6b. You are using a "reference" genome - you are generating a CIGAR string by aligning against a reference, then checking for the shortest CIGAR - this is your measurement of "accuracy" - and you are then striving to minimize this by picking the amount of Illumina data which gives the shortest CIGAR. But my point is that you are *assuming* that your reference gives the best answer, and possibly discarding a longer CIGAR that reveals more of the ground truth, but has a higher disagreement with the reference you are using. Essentially you have developed a method on an unknown and have not tested it on a "known" set. Is it not possible to demonstrate that this error-correction works using a synthetic set that matches or doesn't match a reference to known degrees?

We understand the reviewer's concern but would note that the potential limitation does not apply equally in all contexts. In this case, the HSV-1 reference genome (strain 17) and the experimental strain used (Patton GFP-Us11) differ by less than 850 nt across their ~152kb genomes (Pourchet et al 2017, PMID: 28957690) with almost all differences being SNPs rather than indels. We do agree with the reviewer however that in other systems (e.g. human cytomegalovirus) this could be a problem and acknowledge that our approach, while highly successful in this specific case, would require additional validation were it to be applied to other systems. To this end we have modified the abstract to remove mention of error-correction and the discussion (lines 363) to better reflect the need to evaluate this approach in other systems before use.

As requested, I also read through Reviewer 1's comments, responses to that:

We thank the reviewer for this additional effort.

In response to the first set of comments focusing on read through/inter-transcript splicing - in my opinion the qPCR is validating, but text should be added pointing out the low level of this read-through.

This is a fair point and we have added text to address this on line 283

Importantly, the authors provide evidence of the read-through transcript's translation product via an SDS-PAGE analysis, and as long as presented with appropriate caveats in the discussion (noting the lack of replication/small N), it is reasonable to present these results. Their analysis finding this read-through in the Illumina data generated by Rutkowski et al further validates these results.

We have noted in the legend for Figure 7 (line 688) that the image shown is representative of three replicates.

In summary, we thank all three reviewers and the editor for their thoughtful feedback, the manuscript is much improved because of your expert input.